# RL Squeezes, SFT Expands:
# A Comparative Study of Reasoning LLMs

**Kohsei Matsutani**[*]    **Shota Takashiro**    **Gouki Minegishi**
**Takeshi Kojima**    **Yusuke Iwasawa**    **Yutaka Matsuo**
The University of Tokyo
[*]`kohsei.matsutani@weblab.t.u-tokyo.ac.jp`

## Abstract

Large language models (LLMs) are typically trained by reinforcement learning (RL) with verifiable rewards (RLVR) and supervised fine-tuning (SFT) on reasoning traces to improve their reasoning abilities. However, how these methods shape reasoning capabilities remains largely elusive. Going beyond an accuracy-based investigation of how these two components sculpt the reasoning process, this paper introduces a novel analysis framework that quantifies reasoning paths and captures their qualitative changes under each training process (with models of 1.5B, 7B, and 14B parameters on mathematical and code domains). Specifically, we investigate the reasoning process at two levels of granularity: the trajectory-level, which examines complete reasoning outputs, and the step-level, which analyzes reasoning graphs whose nodes correspond to individual reasoning steps. Notably, clustering of unique reasoning trajectories shows complementary effects: RL compresses incorrect trajectories, whereas SFT expands correct ones. Step-level analysis reveals that RL steepens (about 2.5 times), while SFT flattens (reduced to about one-third), the decay rates of node visitation frequency, degree, and betweenness centrality distributions in the reasoning graph. This indicates that RL concentrates reasoning functionality into a small subset of steps, while SFT homogenizes it across many steps. Furthermore, by evaluating the reasoning graph topologies from multiple perspectives, we delineate the shared and distinct characteristics of RL and SFT. Our work presents a novel reasoning path perspective that explains why the current best practice of two-stage training, with SFT followed by RL, is successful, and offers practical implications for data construction and more efficient learning approaches.

## 1 Introduction

Following the advent of OpenAI-o1 (Jaech et al., 2024) and the open-sourcing of DeepSeek-R1 (Guo et al., 2025), post-training for enhancing reasoning abilities to solve complicated logical tasks, including mathematical problems, has seen a surge of interest. Two primary learning methods are adopted for reasoning post-training: Supervised Fine-Tuning (SFT)[1], where the policy is trained to imitate teacher policies by maximizing log-likelihood using supervision signals from human annotations or strong teacher models (Ye et al., 2025; Muennighoff et al., 2025; Guha et al., 2025); and Reinforcement Learning (RL)[2], which maximizes expected rewards to optimize the probability of producing correct solutions in verifiable tasks (Jaech et al., 2024; Guo et al., 2025).

It has been suggested that RL with verifiable rewards (RLVR) in LLMs simply incentivizes pre-existing capabilities of the base model (Base model) (Liu et al., 2025d; Zhao et al., 2025a; Shah et al., 2025; Gandhi et al., 2025) since it performs Chain-of-Thought (Wei et al., 2022) in vast vocabulary spaces within the constraints of the Base model's prior. Recently, Yue et al. (2025) investigated the *Pass@k* metric (Chen et al., 2021; Song et al., 2025b; Dang et al., 2025; Wen et al.,

---

[1]In this paper, we denote SFT as supervised fine-tuning on reasoning traces generated by reasoning LLMs such as DeepSeek-R1 (Guo et al., 2025) and Gemini-thinking (Comanici et al., 2025).

[2]In this paper, we denote RL as practical policy-gradient methods such as GRPO (Shao et al., 2024).

2025; Wu et al., 2025), which measures the probability that at least one correct solution is found when sampling $k$ independent solutions from the model (i.e., Best-of-$k$). They showed that, as $k$ increases, Base model's *Pass@k* eventually surpasses that of the RL model trained with RLVR. This observation suggests that Base models already possess the capability to solve problems that RL models can solve. However, these studies primarily evaluate answer accuracy without investigating the underlying reasoning process. Additionally, current state-of-the-art models for mathematics and coding, such as ProRL (Liu et al., 2025b) and AceReason (Chen et al., 2025d; Liu et al., 2025e), apply RL starting from DeepSeek-R1 (Guo et al., 2025) distillation model checkpoints, essentially conducting two-stage training with SFT followed by RL (SFT+RL models). DeepSeek-R1 (Guo et al., 2025) also features cold-start integration.

Yet, various SFT+RL training approaches are currently developed through trial-and-error without grasping the distinct roles of RL (reinforcement) and SFT (imitation). An important question to ask is then, *"how do RL and SFT shape the reasoning process beyond accuracy measurements?"*

In this paper, we systematically dive into reasoning process at two granularities (Figure 1): (1) *trajectory-level*, where entire thinking generations are regarded as single trajectory, and (2) *step-level*, where each node (vertex) in the latent space graph (hereafter referred to as the reasoning graph) represents a logical expression (i.e., a sentence), such as a problem setup, a calculation, or a verification.

For *trajectory-level* analysis, we sample multiple outputs from Base, RL, SFT, and SFT+RL models[3], then identify unique trajectories by applying clustering to group similar ones. We find that RL decreases the number of unique incorrect trajectories, whether starting from Base or SFT models, whereas SFT increases the number of unique correct trajectories, suggesting that RL compresses incorrect trajectories while SFT expands correct ones. We also note that SFT alone preserves incorrect trajectories. These results justify the two-stage approach of creating correct trajectories with SFT followed by suppressing incorrect paths with RL. Additionally, RL consistently reduces correct trajectories, which provides an explanation for why Base model's *Pass@k* converges with that of the RL model at large $k$.

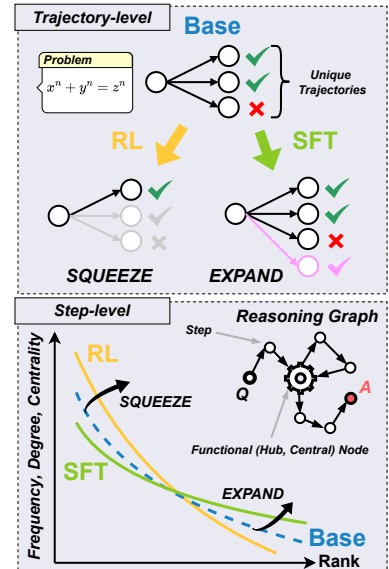

Figure 1: **Overview of our analysis.** (Top) RL compresses incorrect trajectories, and SFT expands correct trajectories. (Bottom) RL concentrates functionality (e.g., hubs) in a small number of steps, and SFT distributes functionality more uniformly across many steps.

At the *step-level*, we construct reasoning graphs by segmenting model outputs into sentences, generating their embeddings, and clustering these representations to define nodes in sentence space. We observe that rank plots for node *visitation frequency*, *degree*, and *betweenness centrality* in reasoning graphs follow exponential laws. Remarkably, analysis of their decay rates reveals that RL elevates the decay rate, whereas SFT degrades it, suggesting that RL not only compresses the graph but also consolidates functionality (e.g., hubs) into fewer nodes (steps).

We further investigated the reasoning graph structure through global and local topological metrics. We identified both shared and distinct patterns in how RL and SFT modify reasoning graph topologies. Both RL and SFT convert local acyclic reasoning graph structures to cyclic ones, resulting in similar subgraph proportions. However, RL transforms the community-structured reasoning graphs of Base models into hub-centralized graphs, while SFT, using traces from a strong teacher, weakens community boundaries to form globally connected graphs.

Our contributions are summarized as follows:

---

[3]Throughout, Base model refers to the model immediately after pretraining, RL model to the Base model after RLVR, SFT model to the Base model after SFT (distillation), and SFT+RL model to the SFT model further trained with RL.

- *Trajectory-level* analysis confirms that RL compresses incorrect trajectories while SFT expands correct ones, highlighting why the two-stage approach (SFT then RL) is effective.

- *Step-level* analysis uncovers that RL also consolidates reasoning graph functionality into fewer steps, whereas SFT expands it across diverse steps. Moreover, through topological metrics, we demonstrate that, while both RL and SFT generate local cyclic structures, they produce distinct global topologies.

- At both *trajectory* and *step-level* analysis, we provided empirical support that RL squeezes and SFT expands the reasoning process. Our findings interpret why existing post-training recipes work and suggest directions for developing new training methods and for data curation.

The code is available at `kohseim/rl_squeezes_sft_expands`.

## 2 RELATED WORK

**RL for LLM.** Research on RLVR has explored how it introduces novel reasoning abilities to LLMs. Yue et al. (2025); Song et al. (2025b); Dang et al. (2025); Wen et al. (2025) argues that RLVR merely elicits existing base capabilities rather than developing new ones, as evidenced by *Pass@k* metrics. Wu et al. (2025) demonstrates theoretically that RLVR cannot exceed the support of Base model. Furthermore, advanced reasoning abilities, such as backtracking and verification (Gandhi et al., 2024; 2025), are amplified only when Base models already possess them (Liu et al., 2025d; Zhao et al., 2025a; Shah et al., 2025). While RLVR underperforms with Llama (Grattafiori et al., 2024) compared to Qwen (Qwen et al., 2025; Yang et al., 2025), mid-training on mathematical domains is crucial (Wang et al., 2025d). Additionally, research explores self-improvement (Huang et al., 2023; Pang et al., 2024; Huang et al., 2025) using iterative internal rewards (Shao et al., 2025; Zhou et al., 2025; Zhao et al., 2025b; Prabhudesai et al., 2025; Cheng et al., 2025; Chandak et al., 2025) such as confidence measures rather than verifiable rewards. It is argued that biased policy gradients can substantially sharpen distributions even with random rewards (Oertell et al., 2025).

**RL vs SFT.** The two dominant paradigms for post-training reasoning LLMs are SFT and RL. Previous works analyzed from the perspective of transfer ability (Han et al., 2025; Li et al., 2025; Chu et al., 2025) and demonstrate that SFT tends to memorize, whereas RL generalizes (Chu et al., 2025), with RL exhibiting superior retention due to negative samples (Lai et al., 2025) and its inherent on-policy nature (Shenfeld et al., 2025). Chen et al. (2025a) observed SFT pseudo-reasoning interferes with RL training in VLMs, and Setlur et al. (2025a) showed the superiority of verifier-based RL under anti-concentration and heterogeneous conditions. Furthermore, several integrated approaches improve performance by combining SFT and RL (Ma et al., 2025; Chen et al., 2025b; Liu et al., 2025c; Yoshihara et al., 2025; Chen et al., 2025c).

**Analysis of Reasoning Behaviors.** Bogdan et al. (2025) analyzed reasoning steps in mathematical domains, Qin et al. (2025) examines plan-execute-verify paradigms, and Liu et al. (2025a) leverages unary "try again" feedback. Liang et al. (2025); Cheng et al. (2025) investigates thinking tokens, with particular attention to overthinking phenomena (Sui et al., 2025) and *aha moments* (Guo et al., 2025) such as "Wait" tokens (Wang et al., 2025a; Ding et al., 2025). Furthermore, studies explore steering vectors (Venhoff et al., 2025), and examine the exploration in reasoning LLMs (Lu et al., 2025; Shojaee et al., 2025). Others focus on the locality structure in the vocabulary space (Prystawski et al., 2023; Kim et al., 2025; Minegishi et al., 2025).

Our work extends beyond outcome-based *Pass@k* metrics to examine how RL and SFT fundamentally shape the reasoning processes of LLMs, offering a novel perspective on the formation of reasoning behavior.

## 3 TRAJECTORY-LEVEL ANALYSIS

We now investigate how RL and SFT fundamentally reshape reasoning trajectories (paths) by analyzing their distinct effects on unique paths. See Appendix C.1 for our problem formulation.

Table 1: **Comparison of Model Variants.** We evaluate Base, RL, SFT, and SFT + RL models across three sizes, 1.5B, 7B, and 14B. See Appendix B.1 for detailed model specifications.

|       | Base Model | RL Model | SFT Model | SFT + RL Model |
|-------|------------|----------|-----------|----------------|
| **1.5B** | Qwen2.5-Math-1.5B | Qwen2.5-Math-1.5B-Oat-Zero | DeepSeek-R1-Distill-Qwen-1.5B | Nemotron-Research-Reasoning-Qwen-1.5B |
| **7B** | Qwen2.5-Math-7B | Qwen2.5-Math-7B-Oat-Zero | DeepSeek-R1-Distill-Qwen-7B | AceReason-Nemotron-7B |
| **14B** | Qwen2.5-14B | Qwen-2.5-14B-SimpleRL-Zoo | DeepSeek-R1-Distill-Qwen-14B | AceReason-Nemotron-14B |

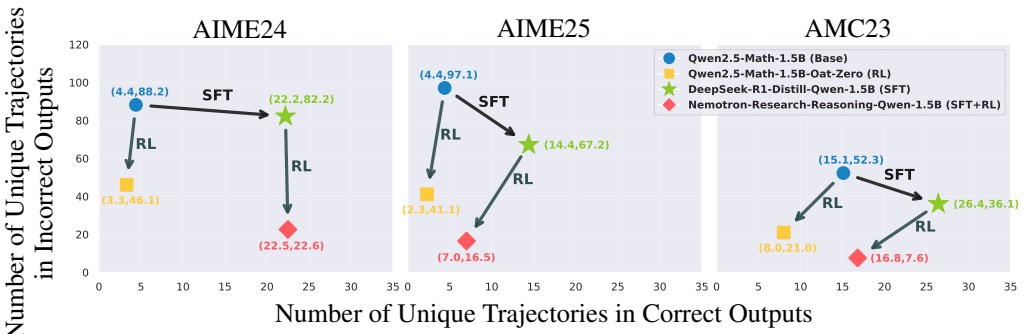

Figure 2: **Effect of RL and SFT on the Number of Unique Trajectories.** The x-axis represents the number of correct clusters and the y-axis represents the number of incorrect clusters for trajectories before and after training of 1.5B models in Table 1. Plot shows the average across samples. See Appendix C.3 for complete results and for additional results Appendix C.4.

## 3.1 CHARACTERIZING UNIQUE REASONING TRAJECTORIES

We comprehensively study Base, SFT, RL, and SFT+RL models in Table 1 on AIME24, AIME25, and AMC23. (See Figure 12 for accuracy comparisons). We also studied the 7B models in Table 1 on HumanEval (Chen et al., 2021). For each problem and model, we generate $M = 256$ samples using a `temperature` of 0.6, `top_p` of 0.95, and a `response_length` of 16000. Meticulous attention must be paid to implementation details regarding prompt templates and response length. Please refer to Appendix B.2 for details. We report *Pass@k* results in Figure 12. These samples comprise both $M_+$ correct trajectories and $M_-$ incorrect trajectories. To estimate the number of unique trajectories, we compute pairwise similarities between the sampled outputs and apply hierarchical clustering based on thresholds. The similarity between two reasoning trajectories $\pi^i$ and $\pi^j$ is measured using the chrF (Popović, 2015):

$$\text{chrF}_\beta = (1 + \beta^2)\frac{\text{CHRP} \cdot \text{CHRR}}{\beta^2 \cdot \text{CHRP} + \text{CHRR}},$$

where CHRR is the proportion of character n-grams in the hypothesis that also occur in the reference and CHRR is the proportion of character n-grams in the reference that also occur in the hypothesis. Compared to BLEU (Papineni et al., 2002), which is based on word-level n-grams, chrF uses character-level n-grams and better captures semantic similarity under morphological variation (e.g., "add" vs. "adding"). See Appendix E for representative examples of trajectories.

Given the verifiable reward, we split a set of $M$ trajectories into the correct set and the incorrect set. For each subset, we construct a similarity matrix $S_+ \in \mathbb{R}^{M_+ \times M_+}$ and $S_- \in \mathbb{R}^{M_- \times M_-}$, where each entry $s_{i,j} = \left(\text{chrF}_\beta(\pi^i, \pi^j) + \text{chrF}_\beta(\pi^j, \pi^i)\right)/2$. The corresponding distances are then defined as $d_{i,j} = 1 - s_{i,j}$, yielding $D_+$ and $D_-$. Since chrF is not an embedding-based metric in Euclidean space, we employ UPGMA (Unweighted Pair Group Method with Arithmetic Mean) (Sokal et al.,

1958) for hierarchical clustering rather than Ward's Method (Ward, 1963) or Centroid Linkage. We use a similarity threshold of 60 to cut the dendrograms and report the resulting number of clusters for the correct and incorrect sets. Figure 2 plots the number of correct clusters on the horizontal axis and incorrect clusters on the vertical axis. The overall similarity distribution is shown in Appendix C.6, and results obtained with BLEU and under different thresholds are provided in Appendix C.5.

## 3.2 RL Squeezes and SFT Expands Unique Reasoning Trajectories

As shown in Figure 2, applying RL from either Base model or SFT model dramatically reduces the number of incorrect trajectories. This indicates that RL enhances *Pass@1* through probability mass redistribution. This aligns with theoretical predictions of empirical support shrinkage (Wu et al., 2025) and diversity collapse (Dang et al., 2025). Crucially, we also observe a reduced number of correct trajectories, explaining why Base models outperform RL models in *Pass@k* at large $k$ (Yue et al., 2025). In contrast, applying SFT to the Base model increases the number of correct trajectories, showing that SFT teaches new solution strategies absent in the Base model. Yet, SFT preserves incorrect trajectories with non-negligible probability mass, potentially improving *Pass@k* but not guaranteeing *Pass@1* gains. Finally, the two-stage SFT+RL procedure demonstrates complementary mechanisms: SFT expands correct trajectories while subsequent RL compresses incorrect trajectories. This combination, which acquires new solution paths through SFT and removes incorrect paths through RL, maximizes *Pass@1* performance. These findings substantiate the state-of-the-art training of SFT followed by RL (Liu et al., 2025b; Chen et al., 2025d; Liu et al., 2025e) from a reasoning trajectory perspective. We obtained consistent results across other models, including the Llama family (Grattafiori et al., 2024), as detailed in Appendix C.4. Consistent results were also obtained on the code domain using HumanEval as shown in Figure 3. See Appendix C.8 for details.

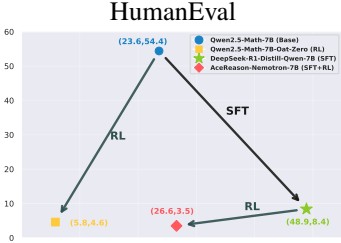

Figure 3: **Effect of RL and SFT on the Number of Unique Trajectories.** The x-axis represents the number of correct clusters and the y-axis represents the number of incorrect clusters for trajectories before and after training of 7B models in Table 1.

## 4 Step-Level Analysis

In step-level analysis, we examine how SFT and RL affect reasoning at a more detailed granularity than trajectory-level analysis. We investigate the reasoning graphs from two perspectives: profiling the global structure (Section 4.2) and capturing the local structure (Section 4.3).

## 4.1 Constructing Reasoning Graph

Consider an evaluation dataset $\mathcal{D} = \{x_n\}_{n=1}^N$ with $N$ problems. Given any input $x \in \mathcal{D}$, we sample $M$ independent responses. Let each model response $\pi_m^l$ (for response sample $m \in [M]$) be segmented into sentences: $\pi_m^l = (r_{m,1}^l, r_{m,2}^l, \ldots, r_{m,T_m^l}^l)$, where $l \in \{\text{Base}, \text{SFT}, \text{RL}, \text{SFT+RL}\}$ indexes the model variant and $T_m^l$ denotes the number of sentences in response $\pi_m^l$. Each sentence $r_{m,t}^l$ is mapped into a $d$-dimensional vector space via a sentence embedding function, yielding $s_{m,t}^l \in \mathbb{R}^d$. For each problem $x \in \mathcal{D}$, we define the set of sentence embeddings $\mathcal{S}(x) = \{ s_{m,t}^l \mid l \in \{\text{Base}, \text{RL}, \text{SFT}, \text{SFT+RL}\}, m \in [M], t \in [T_m^l] \}$, and collect them across all problems as $\mathcal{S} = \{ (x, s) \mid x \in \mathcal{D}, s \in \mathcal{S}(x) \}$. We perform unsupervised clustering of $\mathcal{S}$ using $K$-means. This yields a partition of $\mathcal{S}$ into $K$ clusters with representative centroids $C = \{c_1, \ldots, c_K\}, \quad c_k \in \mathbb{R}^d$.

We denote each cluster by a node $v_k$, so that the node set is $\mathcal{V} = \{v_1, \ldots, v_K\}$. We define the node set of $\pi_m^l$ as $\mathcal{V}_m^l = \{ v \in \mathcal{V} \mid \exists t : s_{m,t}^l \mapsto v \}$, Each embedding $s_{m,t}^l$ is assigned to a unique node $v_k$. The distance between two nodes $v_i$ and $v_j$ is defined as the Euclidean distance between their centroids: $d(v_i, v_j) = \|c_i - c_j\|_2$. For each response trajectory $\pi_m^l$, we derive a corresponding sequence of node transitions. Consecutive occurrences of the same cluster assignment are merged into a single node to avoid self-loops. This induces a directed edge set $\mathcal{E}_m^l = \{(v_i \rightarrow$

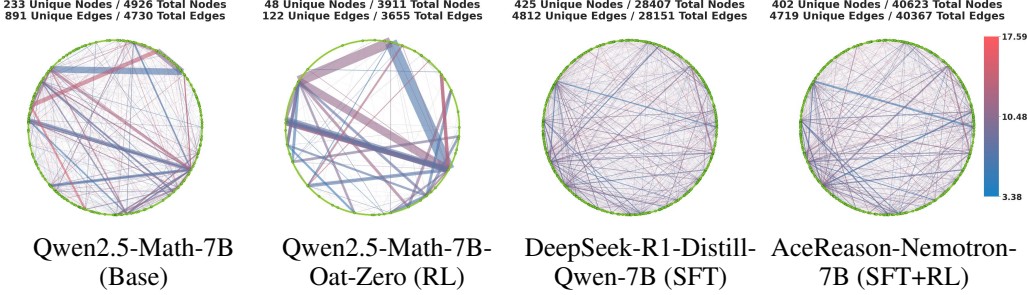

Figure 4: **Reasoning Graph Example.** Reasoning graph for AIME24 Problem #1 built from 256 responses across the 7B models in Table 1. Nodes are arranged sequentially on a circle, consistent across examples. Edge thickness encodes transition frequency, and edge color encodes edge distance. For more examples, please refer to Figure 22 and Figure 23.

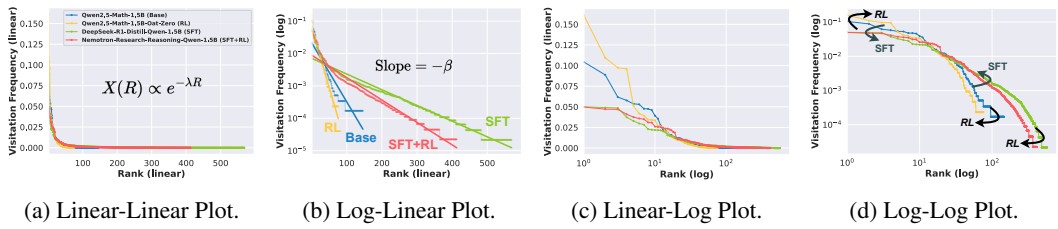

(a) Linear-Linear Plot.     (b) Log-Linear Plot.     (c) Linear-Log Plot.     (d) Log-Log Plot.

Figure 5: *Visitation Frequency* **Rank Plot.** Results from the 1.5B model in Table 1 on AIME24 Problem #1, shown with four combinations of linear/log scales on the x- and y-axes. The x-axis represents the node rank, and the y-axis represents the *Visitation Frequency* at each rank. The rank plot approximately follows an exponential law, showing near-linear behavior on a log-linear scale. See Figure 24 for more examples.

$v_j) \mid v_i, v_j$ are consecutive and distinct cluster assignments in some $\pi_m^l$}. Each edge $(v_i \to v_j)$ is associated with $d(v_i, v_j)$ and the frequency of this transition. Thus, each model LLM $l$ generates a response $\pi_m^l$ that can be represented as a path in the directed graph $\mathcal{G}_m^l = (\mathcal{V}_m^l, \mathcal{E}_m^l)$, where nodes correspond to clustered semantic units and edge weights reflect their inter-cluster distances.

In our implementation, we employ BGE-large-en-v1.5 (Xiao et al., 2024) as the sentence embedding, where $d = 1024$, set $M = 256$ and $K = 2000$. We conduct experiments for models in Table 1 on AIME24, AIME25, and AMC23 and for the 7B models in Table 1 on HumanEval. For more details on the implementation, see Appendix D.1. We conduct ablations of the reasoning graph construction for the 7B models in Table 1, varying (i) the number of clusters from our default $K = 2000$ to $K = 1000, 3000$, (ii) the distance metric from Euclidean (L2) distance to cosine distance, and (iii) the sentence encoder from BGE-large-en-v1.5 to GTE-base-en-v1.5 (Zhang et al., 2024) with $d = 768$. Details are provided in Appendix D.4. While our approach builds on Wang et al. (2024); Minegishi et al. (2025), who averaged token representations extracted from each Transformer block within chunks, and performed clustering on hidden states for a single model, we instead embed sentences into a shared embedding space and cluster their vector representations jointly across four models. Whereas using each model's internal representations would result in graphs that live in different representation spaces, constructing graphs in this shared sentence embedding space enables direct comparison of the graph properties induced by different models.

## 4.2 GLOBAL REASONING GRAPH STRUCTURE

The graph visualizations are presented in Figure 4. We observe that RL strengthens some edges while pruning others, whereas SFT creates new connections (see Appendix D.2 for node disparities). For each model and each problem $x$, we consider a weakly connected reasoning graph: $\mathcal{G}^l = \bigcup_{m=1}^M \mathcal{G}_m^l = (\mathcal{V}^l, \mathcal{E}^l)$.

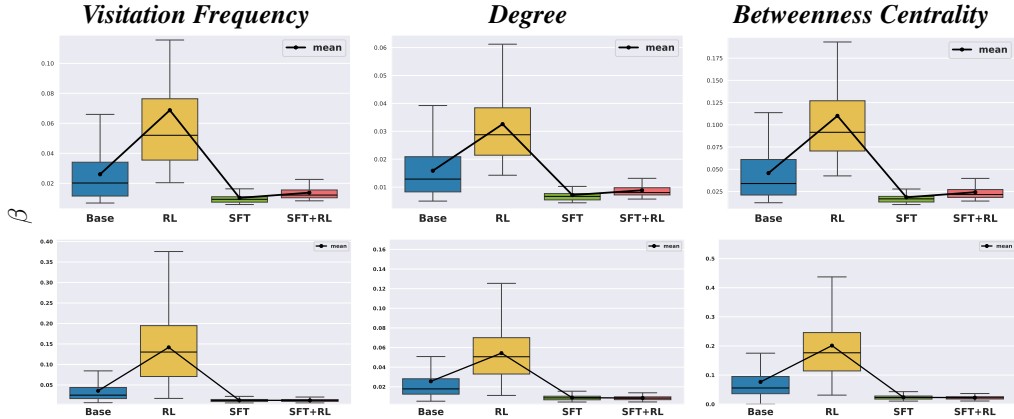

Figure 6: **Exponential Decay Rate for *Visitation Frequency*, *Degree*, and *Betweenness Centrality*.** Box plots show the estimated exponential decay rate $\beta$ for the **(Top)**, computed across all problems in AIME24, AIME25, and AMC23 for the 1.5B models in Table 1; and for the **(Bottom)**, computed across all problems in HumanEval for the 7B models in Table 1. See Figure 26 for complete results.

**Estimating Exponential Decay Rate.**    We investigate how RL and SFT modulate the structure and function (Newman, 2003) of complex reasoning graphs by examining the distributional properties of node visitation frequency (*visitation frequency*), node degree (*degree*), and *betweenness centrality* (Freeman, 1977) within the graphs. For each $\mathcal{G}^l$, *visitation frequency* is given by $\frac{n(v)}{\sum_{u \in \mathcal{V}} n(u)}$ where $n(u)$ is the number of node visit and *degree* is given by $|\{\, u \in \mathcal{V}^l \;\mid\; (v \to u) \in \mathcal{E}^l \;\lor\; (u \to v) \in \mathcal{E}^l \,\}|$. *Betweenness centrality* is defined as $\frac{1}{(|\mathcal{V}^l|-1)(|\mathcal{V}^l|-2)} \sum_{s \neq v \neq t} \frac{\sigma_{st}(v)}{\sigma_{st}}$ where $\sigma_{st}$ is the number of shortest paths (determined by edge count) from node $s$ to node $t$ and $\sigma_{st}(v)$ is the number of those shortest paths that pass through node $v$ $(v \neq s, t)$. *Betweenness centrality* measures the importance of the reasoning step in mediating the shortest connections in the graph.

For each graph $\mathcal{G}^l$, we present the rank plots of *visitation frequency* in Figure 5. We observe that the rank plots of *visitation frequency*, *degree*, and *betweenness centrality* approximately exhibit exponential decay (see Figure 24 for additional rank plots), and hence follow an exponential law. This corresponds to approximately linear decay in log-linear rank plots. As can be seen in Figure 5b, the plots (models) exhibit markedly different decay rates (i.e., slopes). We therefore investigate the magnitude of this exponential decay. Suppose that the associated value

$$X(R) \propto e^{-\lambda R},$$

where $R$ denotes the rank of a node and $\lambda$ governs the rate of decay. We estimate **exponential decay rate** $\beta = \frac{\lambda}{\log 10}$ by linear regression, which is given by $\log_{10} X(R) = \alpha - \beta R + \epsilon_R$, where $\alpha$ is an intercept and $\epsilon_R$ denotes deviations. Figure 5b shows an example (additional examples in Figure 25).

**RL Squeezes and SFT Expands Graph Functionalities.**    We estimated the exponential decay rate $\beta$ for the reasoning graphs $\mathcal{G}^l$ across all problems $x$ in AIME24, AIME25, and AMC23. The results are presented as box plots in Figure 6 (full results in Figure 26).

The transition from Base through RL reveals a pronounced structural reorganization, characterized by a marked increase in $\beta$. This reflects that high-rank nodes exhibit elevated *visitation frequency*, *degree*, and *betweenness centrality* while low-rank nodes exhibit reduced values of these measures. This suggests that RL consolidates key graph functions (frequency, degree, and centrality) into fewer nodes. In stark contrast, SFT reveals an inverse pattern with reduced $\beta$. High-rank nodes display decreased *visitation frequency*, *degree*, and *betweenness centrality*, whereas low-rank nodes show increased levels of these measures. This divergent behavior indicates that RL aggregates functional steps (e.g., hub, central nodes) in the reasoning graph into a small number of steps (nodes), whereas SFT, conversely, diversifies them across many steps. We also obtained results on HumanEval that align with these results on mathematical domains, as shown in Figure 6. See Figure 37 for details.

**Edge Density** — **Normalized Clustering Coefficient** — **Assortativity** — **Modularity** — **Freeman Centralization** — **Normalized Average Path Length** — **Global Efficiency** — **Algebraic Connectivity**

Figure 8: **Comparison of eight graph metrics across Base, RL, SFT, and SFT+RL models.** Values are averaged across different model sizes in Table 1 and three datasets, AIME24, AIME25, and AMC23. For details on the eight metrics, see Appendix D.3. See Figure 31 for results by model size.

In our construction of reasoning graphs, nodes are defined by clustering sentences, which tends to produce graphs with high edge density. To address this, we apply graph sparsification by retaining, for each node, the top-10 or top-20 edges with the smallest Euclidean norm and then estimate the exponential decay rate $\beta$, as detailed in Appendix D.5. We obtain consistent results where RL decreases and increases SFT increases $\beta$, as shown in Figure 33.

**Profiling Global Structure.** Next, we profile the global structure of reasoning graphs $\mathcal{G}^l$ through eight topology metrics. We present the edge density, clustering coefficient normalized by the random graph (Watts & Strogatz, 1998), assortativity (Newman, 2002), modularity (Girvan & Newman, 2002), Freeman centralization (Freeman, 1978), average path length normalized by the random graph (Watts & Strogatz, 1998), global efficiency (Latora & Marchiori, 2001), and algebraic connectivity (Fiedler, 1973) of each model's reasoning graph in Figure 8. For detailed descriptions of each metric, see Appendix D.3. As shown in Figure 8, the reasoning graph of Base model exhibits notably high modularity (Figure 7), low global efficiency, and low algebraic connectivity. This indi-

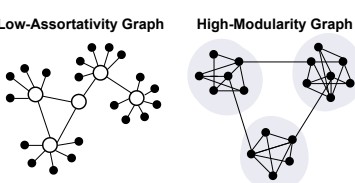

Figure 7: **Illustration of the Low-Assortativity Graph (Left) and High-Modularity Graph (Right).**

cates that the nodes are organized into distinct communities (clusters) with weak inter-community connections. Consequently, the reduced robustness and poor reachability efficiency limit the model's ability to fully explore the reasoning graph. However, after RL from Base model, we observe high edge density, low clustering coefficient, low assortativity (Figure 7), and high Freeman centralization. This characterizes a graph dominated by a small number of high-degree hubs densely connected to peripheral nodes. This can be interpreted as the Base model's reasoning graph being squeezed into a structure that enables efficient traversal through a small set of hub nodes. We also obtained results on HumanEval that align with these results on mathematical domains, as shown in Figure 38. Finally, SFT and SFT+RL models exhibit low modularity, high global efficiency, and high algebraic connectivity. This reveals a reasoning graph characterized by high robustness and superior reachability efficiency, without distinct community structures.

We observe that global efficiency and algebraic connectivity are positively correlated with Pass@1/Pass@$k$, whereas modularity is negatively correlated. This suggests that these metrics relate to the model's ability to effectively explore the solution space and reach the correct answer in a single attempt. Details are provided in Appendix D.3. Furthermore, graph sparsification in Appendix D.5 exhibited the same trend in the changes of the graph metric under RL and SFT, as shown in Figure 34.

## 4.3 LOCAL REASONING GRAPH STRUCTURE

**Capturing Local Structure with Graphlets.** We now turn our attention to local structural differences in reasoning graphs, we employ graphlet analysis (Milo et al., 2004; Pržulj et al., 2004), which examines small, connected, nonisomorphic induced subgraphs (see Appendix D.3 for more details.).

We count the 4-node graphlet subgraphs shown in Figure 9a in each model's graph. Figure 9b shows that with RL as well as SFT, we observe a decrease in the proportion of acyclic subgraphs, such as G3 and G4, while cyclic structures like G7 and G8 increase. This indicates that RL introduces local cyclic structures, reflecting backtracking and verification (Gandhi et al., 2025), into the reasoning graph. Moreover, compared to Base model, the RL, SFT, and SFT+RL models all exhibit similar 4-node graphlet proportions.

However, as shown in Figure 12, there are significant performance gaps between the RL model and the SFT/SFT+RL models. This suggests that local structure alone cannot fully explain reasoning performance and global structure, as discussed in Section 4.2, seems to play a crucial role.

(a) **4-node Graphlets (G3–G8).**

(b) **Proportion of Graphlets.**

Figure 9: **Graphlets (G3–G8).** (a) 4-node graphlet motifs and (b) their proportions averaged across all models in Table 1 and datasets (AIME24, AIME25, AMC23). Arrows indicate the change in proportion after RL.

## 5 DISCUSSION

In this work, we explored how RL and SFT influence mathematical reasoning through a novel reasoning path perspective, examining both *trajectory* and *step-level* granularities across multiple model sizes and datasets. From Section 3, the practical success of RL from SFT can be explained by expanding correct trajectories and then compressing incorrect ones, which implies improvements in both *Pass@k* and *Pass@1*. Of particular note is that SFT preserves incorrect trajectories, indicating that SFT alone does not guarantee *Pass@1* performance. This finding precisely accounts for the experimental observation of *pseudo reasoning paths* induced by SFT in (Chen et al., 2025a). Additionally, Section 4.2 reveals that this path-squeezing effect by RL concentrates functionalities (e.g., hubs) into fewer nodes (steps). In contrast, SFT homogenizes these functionalities across diverse steps.

**Contrastive Mechanism of RL and SFT.** Recently, Wang et al. (2025c;b) observed at the token level that high entropy thinking (forking) tokens drive reasoning, and RL increases their entropy while decreasing the entropy of non-thinking tokens. Similarly, we observed that RL amplifies the difference between steps with high frequency, degree, and centrality, and other steps. Therefore, applying RL only to functional steps (e.g., hub or central steps) could further improve LLM reasoning performance and enable more efficient learning. The empirical finding that SFT memorizes and RL generalizes (Chu et al., 2025) may also be related to RL's aggregation and SFT's distribution of reasoning functionalities. In addition, several studies have investigated RL with exploration bonuses (Cheng et al., 2025; Setlur et al., 2025b; Zheng et al., 2025; Chen et al., 2025e; Song et al., 2025a). It would be valuable to analyze whether these approaches merely prevent collapse due to excessive squeezing of the reasoning graph, or whether they truly expand it in a manner analogous to SFT.

**Reasoning Graph Structure.** We observed that RL weakens community structure and promotes efficient transitions in reasoning graphs. This structural shift by RL mirrors the promotion of inter-cluster transitions in the community structure (Prystawski et al., 2023; Wang et al., 2024; Kim et al., 2025) induced by pre-training on a large language corpus. In SFT data curation for reasoning, Muennighoff et al. (2025) heuristically count "wait" tokens, (Gandhi et al., 2025) prime cognitive behaviors, and Ye et al. (2025) assess step-by-step clarity in the reasoning traces. Our finding that both RL and SFT increase local cyclic structures (Section 4.3) indicates that step-level reasoning behavior is applicable to dataset curation for efficient reasoning. Leveraging the insight that graph structures facilitating traversal without high modularity are critical for reasoning ability, one promising direction is to incorporate graph metrics (e.g., hub and central nodes) as process rewards in RL.

Our experiments mainly focused on verifiable and competitive mathematical and code domains to evaluate strong reasoning LLMs and Because the DeepSeek-R1-Distill family is performed SFT with multiple responses per problem (Guo et al., 2025), we additionally perform SFT on s1k-1.1 dataset (Muennighoff et al., 2025) using a single response per problem. For the 1.5B models in Table 1, we find that our results are hold across both settings, at both the trajectory level (Appendix C.7)

and the step level (Appendix D.6). However, we focused on principled algorithmic differences between RL and SFT without controlling for differences in the training datasets. Investigating their effects on reasoning paths under distribution shift presents intriguing directions, with Han et al. (2025); Li et al. (2025); Chu et al. (2025); Shenfeld et al. (2025) pursuing similar investigations on generalization and forgetting.

ACKNOWLEDGMENTS

This work was supported by the AI Symbiotic Future Society Program at The University of Tokyo.

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

## A    LLM USAGE

We used LLMs for writing, such as grammar correction and rephrasing, coding, and debugging. All generated contents are reviewed and validated by the authors.

## B    SAMPLE GENERATION

### B.1    MODELS

Table 2: **Comparison of Model Variants.** Summary of experimental models used in this study.

| Base Model | RL Model | SFT Model | SFT + RL Model |
|---|---|---|---|
| Qwen2.5-Math-1.5B (Yang et al., 2024) | Qwen2.5-Math-1.5B-Oat-Zero (Liu et al., 2025d) | DeepSeek-R1-Distill-Qwen-1.5B (Guo et al., 2025) | Nemotron-Research-Reasoning-Qwen-1.5B (Liu et al., 2025b) |
| Qwen2.5-Math-7B (Yang et al., 2024) | Qwen2.5-Math-7B-Oat-Zero (Liu et al., 2025d) | DeepSeek-R1-Distill-Qwen-7B (Guo et al., 2025) | AceReason-Nemotron-7B (Chen et al., 2025d) |
| Qwen2.5-14B (Qwen et al., 2025) | Qwen-2.5-14B-SimpleRL-Zoo (Zeng et al., 2025) | DeepSeek-R1-Distill-Qwen-14B (Guo et al., 2025) | AceReason-Nemotron-14B (Chen et al., 2025d) |
| Qwen2.5-Math-7B (Yang et al., 2024) | | | AceReason-Nemotron-1.1-7B (Liu et al., 2025e) |
| Qwen2.5-7B (Qwen et al., 2025) | Qwen2.5-7B-SimpleRL-Zoo (Zeng et al., 2025) | | |
| Llama-3.1-8B (Grattafiori et al., 2024) | Llama-3.1-8B-SimpleRL-Zoo (Zeng et al., 2025) | DeepSeek-R1-Distill-Llama-8B (Guo et al., 2025) | |

We conducted evaluation using the models specified in Table 2. Qwen2.5-7B-SimpleRL-Zoo and Qwen-2.5-14B-SimpleRL-Zoo are trained from Qwen2.5-7B and Qwen2.5-14B, respectively, using GRPO (Yang et al., 2024) without format rewards. Qwen2.5-Math-1.5B-Oat-Zero and Qwen2.5-Math-7B-Oat-Zero are RL-trained from Qwen2.5-Math-1.5B and Qwen2.5-Math-7B, respectively, using Dr.GRPO (Liu et al., 2025d), an improved version of GRPO. Dr.GRPO enhances token efficiency by removing the dividing term and regularization term from GRPO. DeepSeek-R1-Distill-Qwen-1.5B, DeepSeek-R1-Distill-Qwen-7B, and DeepSeek-R1-Distill-Qwen-14B are fine-tuned from Qwen2.5-Math-1.5B, Qwen2.5-Math-7B, and Qwen2.5-14B, respectively, using SFT with DeepSeek-R1's distillation data. Nemotron-Research-Reasoning-Qwen-1.5B undergoes prolonged RL training for 2500 steps from DeepSeek-R1-Distill-Qwen-1.5B. AceReason-Nemotron-7B and AceReason-Nemotron-14B are trained from DeepSeek-R1-Distill-Qwen-7B and DeepSeek-R1-Distill-Qwen-14B, respectively, using large-scale RL on mathematics and coding tasks. AceReason-Nemotron-1.1-7B is trained through large-scale curriculated SFT from Qwen2.5-Math-7B, followed by large-scale RL on mathematics and coding tasks. The SFT checkpoint is not publicly available. Llama-3.1-8B-SimpleRL-Zoo is derived from Llama-3.1-8B via RL, and DeepSeek-R1-Distill-Llama-8B is derived via SFT.

### B.2    INFERENCE

**Prompts.**    In the context of LLM reasoning inference, accuracy demonstrates high sensitivity to prompt template design, necessitating careful attention to template selection and construction. The prompt templates employed in our methodology are showd in Figure 10. We applied the Qwen Template to the following models: Qwen2.5-Math-1.5B, Qwen2.5-Math-1.5B-Oat-Zero, Qwen2.5-7B, Qwen2.5-Math-7B, Qwen2.5-7B-SimpleRL-Zoo, Qwen2.5-Math-7B-Oat-Zero, Qwen2.5-14B, and

**Qwen Template**

```
<|im_start|>system
You are a helpful assistant.<|im_end|>
<|im_start|>user
{input}
Please reason step by step, and put your final answer within
↪    \boxed{}.<|im_end|>
<|im_start|>assistant
```

**Nemotron-Qwen Template**

```
<|im_start|>system
You are a helpful and harmless assistant. You should think
↪    step-by-step.<|im_end|>
<|im_start|>user
{input}
Please place your final answer inside \boxed{}.<|im_end|>
<|im_start|>assistant
<think>
```

**R1 Template**

```
<|begin__of__sentence|><|User|>{input}
Please reason step by step, and put your final answer within
↪    \boxed{}.<|Assistant|><think>
```

**Llama Template**

```
Question:
{input}
Answer:
Let's think step by step.
```

Figure 10: Prompt Templates

Qwen-2.5-14B-SimpleRL-Zoo. Although base models have not been fine-tuned with special tokens and are expected to achieve peak performance without templates (Liu et al., 2025d), we employed the Qwen Template to ensure explicit generation of stop tokens and maintain experimental consistency with the conditions in Zeng et al. (2025); Yue et al. (2025). The R1 Template was utilized for DeepSeek-R1-Distill-Qwen-1.5B, Nemotron-Research-Reasoning-Qwen-1.5B, DeepSeek-R1-Distill-Qwen-7B, AceReason-Nemotron-7B, AceReason-Nemotron-1.1-7B, DeepSeek-R1-Distill-Qwen-14B, AceReason-Nemotron-14B, and DeepSeek-R1-Diatill-Llama-8B. Please replace '␁' and '|' in fig. 10 with U+2581 and U+FF5C, respectively. Following Zeng et al. (2025); Yue et al. (2025), we used Llama Template for Llama-3.1-8B and Llama-3.1-8B-SimpleRL-Zoo. For AceReason-Nemotron-1.1-7B, we employed the Nemotron-Qwen Template (Liu et al., 2025e). Of particular note, for the models Qwen2.5-Math-1.5B, Qwen2.5-Math-1.5B-Oat-Zero, DeepSeek-R1-Distill-Qwen-1.5B, and Nemotron-Research-Reasoning-Qwen-1.5B, we identified potential concerns regarding Chinese-English language mixing in the generated outputs. To mitigate this issue, we appended the instruction "Always respond in English only." to the end of each user prompt.

**Parameters.** For both trajectory-level and step-level experiments, we employed sampling with `temperature=0.6` and `top_p=0.95`. Additionally, we set the response length parameter to 16000 tokens. While the source code implementation in Yue et al. (2025) utilized vLLM with `max_model_len=4096`, this configuration constrains the response length. Consequently, it leads to performance degradation for models that generate extended outputs, such as the DeepSeek-R1-Distill and AceReason-Nemotron families. To address this limitation, we in-

creased the `max_model_len` parameter to 16000 tokens. However, for models with architectural constraints of `max_positional_embeddings=4096`, specifically Qwen2.5-Math-1.5B, Qwen2.5-Math-1.5B-Oat-Zero, Qwen2.5-Math-7B, and Qwen2.5-Math-7B-Oat-Zero, we maintained `max_model_len=4096`. Accuracy degradation due to `max_model_len` is presented in Figure 11.

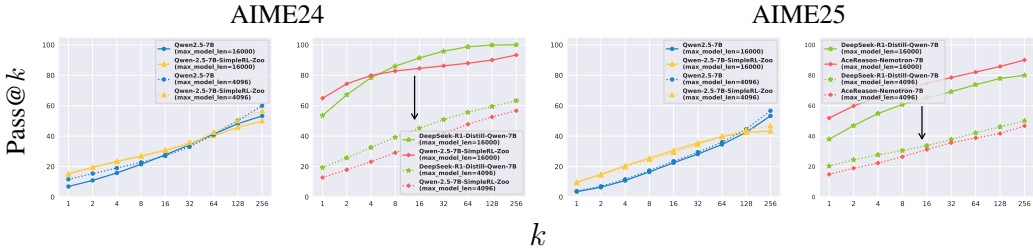

Figure 11: **Effect of `max_model_len` on Pass@$k$.** Misspecification of `max_model_len` causes substantial accuracy degradation in the DeepSeek-R1-Distill-based models, which performs lengthy reasoning.

**Pass@$k$.** We conducted evaluations on AIME24, AIME25, and AMC23 using the same implementation as Yue et al. (2025). For each problem $x_i$ contained in the evaluation dataset $\mathcal{D} = \{x_i\}_{i=1}^n$, we sampled $n$ responses and computed the Pass@$k$ metric for the correct samples $c_i$, which is given by: $\text{pass@}k := \mathbb{E}_{x_i \sim \mathcal{D}} \left[ 1 - \binom{n - c_i}{k} / \binom{n}{k} \right]$. The results for $n = 256$ are shown in Figure 12.

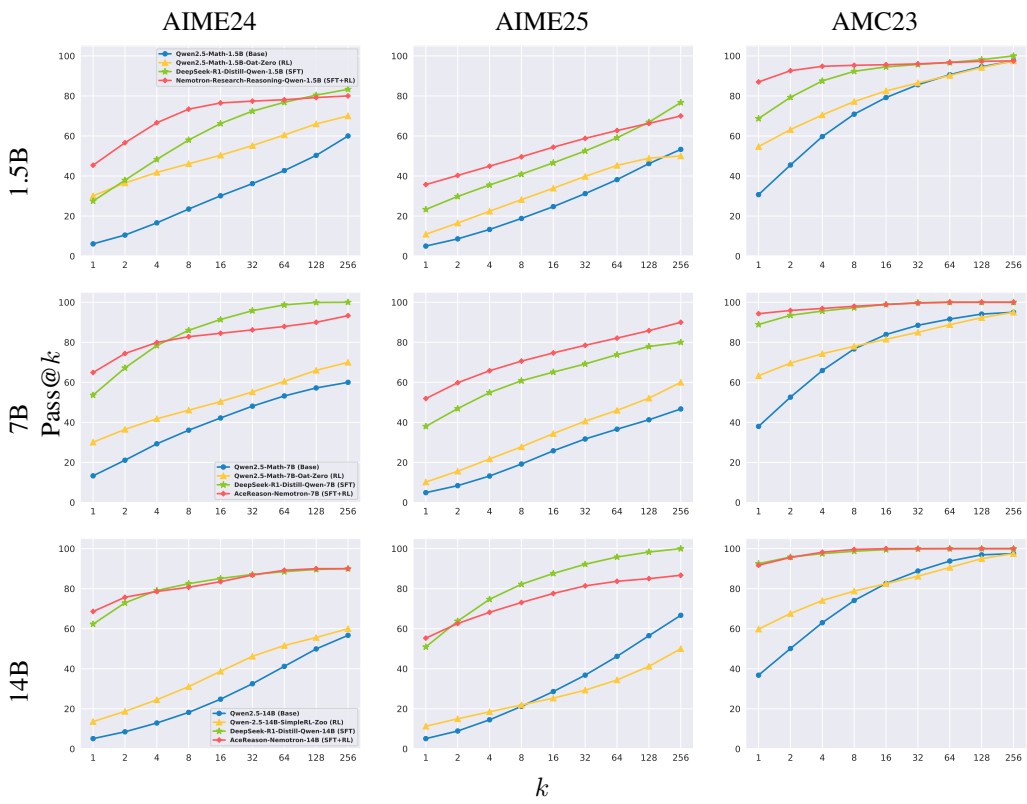

Figure 12: **Pass@$k$ performance curves for Base, SFT, RL, and SFT + RL models.** Models are in the Table 1 and datasets are AIME24, AIME25, and AMC23.

**Response Length.** The mean response length for each model across datasets is presented in Figure 13. Following SFT via distillation from DeepSeek-R1, a notable increase in response length is observed.

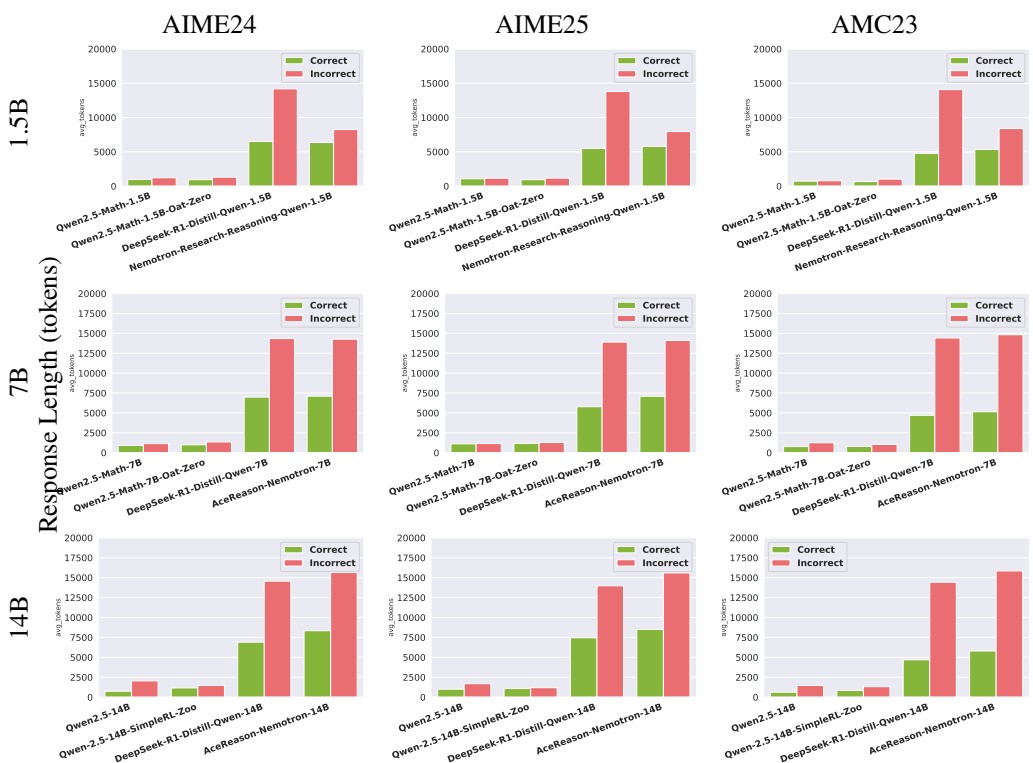

Figure 13: **Comparison of Response Length Across Models and Datasets.** Models are in the Table 1 and datasets are AIME24, AIME25, and AMC23.

# C   TRAJECTORY LEVEL ANALYSIS

## C.1   PROBLEM FORMULATIONS

We consider the LLM parameterized by $\theta$, which defines a probability distribution on discrete reasoning trajectories (paths). Let $\mathcal{X}$ denote the input space of natural language problems, and $\mathcal{Y}$ denote the output vocabulary space. Given an input $x \in \mathcal{X}$, a path $\pi = (y_1, \ldots, y_T)$ is generated with probability $\pi_\theta(\pi \mid x) = \prod_{t=1}^{T} p_\theta(y_t \mid x, y_{<t})$, where $\sum_{\pi \in \Pi(x)} \pi_\theta(\pi \mid x) = 1$. Each path is assigned a binary reward $r(x, \pi) \in \{0, 1\}$, with the set of correct paths denoted by $\Pi_+(x)$ and incorrect paths by $\Pi_-(x) = \Pi(x) \setminus \Pi_+(x)$. The probability of sampling a correct path, corresponding to *Pass@1*, is $p_+(x; \theta) = \sum_{\pi \in \Pi(x)} \pi_\theta(\pi \mid x) r(\pi.x) = \sum_{\pi \in \Pi_+(x)} \pi_\theta(\pi \mid x)$. To achieve the ultimate goal of improving *Pass@1*, SFT and RL are utilized in training reasoning LLMs. SFT optimizes the model to maximize the likelihood of demonstrated trajectories $\pi^*$:

$$\mathcal{L}_{\mathrm{SFT}}(\theta) = -\mathbb{E}_{(x, \pi^*) \sim \mathcal{D}} \left[ \log \pi_\theta(\pi^* \mid x) \right].$$

While RL aims to maximize the expected reward under the model distribution:

$$J(\theta) = \mathbb{E}_{x \sim \mathcal{D}} \, \mathbb{E}_{\pi \sim \pi_\theta(\cdot \mid x)} \left[ r(x, \pi) \right].$$

Put differently,

$$J(\theta) = \mathbb{E}_{x \sim \mathcal{D}} \Big[ \sum_{\pi \in \Pi_+(x)} \pi_\theta(\pi \mid x) \Big] = \mathbb{E}_{x \sim \mathcal{D}} \left[ p_+(x; \theta) \right],$$

which corresponds exactly to maximizing the probability of sampling a correct path (i.e., improving *Pass@1*). Note that, in this research, RL refers to policy-gradient methods such as GRPO (Shao et al., 2024).

In our trajectory-level analysis, we experimentally investigate how SFT and RL affect the LLM's paths by counting the number of unique paths. Let $\mathcal{D} = \{x_n\}_{n=1}^{N}$ be an evaluation dataset consisting of $N$ problems. For each input $x \in \mathcal{D}$, we generate $M$ independent samples from the trajectory distribution $\pi_\theta(\cdot \mid x)$.

The set of trajectories observed in these samples is $\widehat{\Pi}_M(x) = \{\pi \in \Pi(x) : \exists j \leq m \;\; \pi^{(j)} = \pi\}$, where $\pi^{(1)}, \ldots, \pi^{(m)} \sim \pi_\theta(\cdot \mid x)$. This set includes both correct trajectories ($\pi \in \Pi_+(x)$) and incorrect ones ($\pi \in \Pi_-(x)$). Since $\widehat{\Pi}_M(x)$ is obtained by random sampling, not all trajectories in $\Pi(x)$ necessarily appear, but those with a higher probability mass are more likely to occur multiple times within the $M$ samples. We set $M = 256$ in the experiments.

*Pass@k* (Chen et al., 2021; Yue et al., 2025) is the probability that at least one correct solution is found when sampling $k$ independent solutions from the model (i.e., Best-of-$k$), which is given by $Pass@k(x; \theta) = 1 - (1 - p_+(x; \theta))^k$. Yue et al. (2025) found that as $k$ increases, the base model catches up to the RL model in Pass@$k$, indicating that the reasoning paths of the RL model are contained within the base model's sampling distribution. We evaluate the models listed in Table 1, comparing the base model, SFT model, RL model, and SFT + RL model performance. Given their high capabilities, we conduct experiments on challenging mathematical datasets, AIME24, AIME25, and AMC23.

## C.2   IMPLEMENTATION DETAILS

The chrF parameter was set to $\beta = 2$, and for UPGMA clustering, the similarity threshold was set to 60, meaning the distance similarity was calculated at 0.4.

## C.3   EXPERIMENTAL RESULTS

The changes in correct and incorrect paths for the models specified in Table 1 across AIME24, AIME25, and AMC23 are presented in Figure 14. The results demonstrate that RL reduces the number of incorrect paths, while SFT increases the number of correct paths. For the 14B model on AIME25 and AMC23, the AceReason-Nemotron-14B (SFT + RL) model shows minimal changes in path count compared to the SFT model. However, Figure 12 shows that AceReason-Nemotron-14B

does not show performance improvement over the pre-RL DeepSeek-R1-Distill-Qwen-14B on the AIME25 and AMC23 domains, suggesting that the training may not have been successful at this domain.

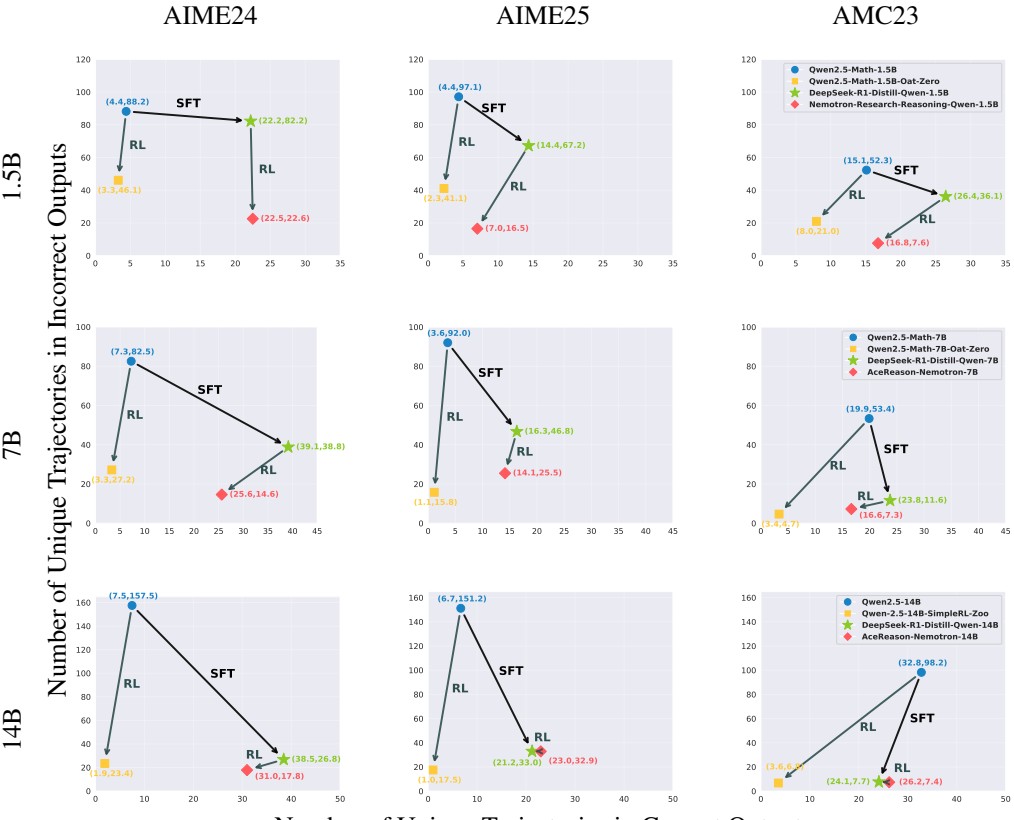

Figure 14: **Effect of RL and SFT on the Number of Unique Trajectories.** The x-axis represents the number of correct clusters and the y-axis represents the number of incorrect clusters for trajectories before and after training of 1.5B, 7B, and 14B models in Table 1

Additionally, in Appendix E, we present examples where RL compresses unique incorrect trajectories, and examples where SFT preserves incorrect trajectories, resulting in different error patterns.

## C.4 MORE EXPERIMENTAL RESULTS

In addition to the models in Table 1, we conducted experiments on Qwen-2.5-7B-SimpleRL-Zoo, AceReason-Nemotron-1.1-7B, Llama-3.1-8B, Llama-3.1-8B-SimpleRL-Zoo, and DeepSeek-R1-Distill-Llama-8B (Model details appear in Table 2). The Pass@$k$ results for each model are presented in Figure 15, while the trajectory-level changes in the number of correct and incorrect paths are detailed in Figure 16. When Qwen2.5-7B undergoes RL training, incorrect paths are substantially compressed. Moreover, AceReason-Nemotron-1.1-7B, after SFT and RL training, expands correct paths while squeezing incorrect paths on AIME24 and AIME25. Conversely, for AMC23, AceReason-Nemotron-1.1-7B exhibits squeezing of both correct and incorrect paths compared to the base model. This phenomenon occurs because, in the AMC23 domain, as shown in Figure 15, Qwen2.5-Math-7B achieves Pass@$k$ performance comparable to AceReason-Nemotron-1.1-7B as $k$ increases, leading to saturation. Finally, we confirm that Llama-3.1-8B also demonstrates squeezing of incorrect paths and expansion of correct paths following RL training.

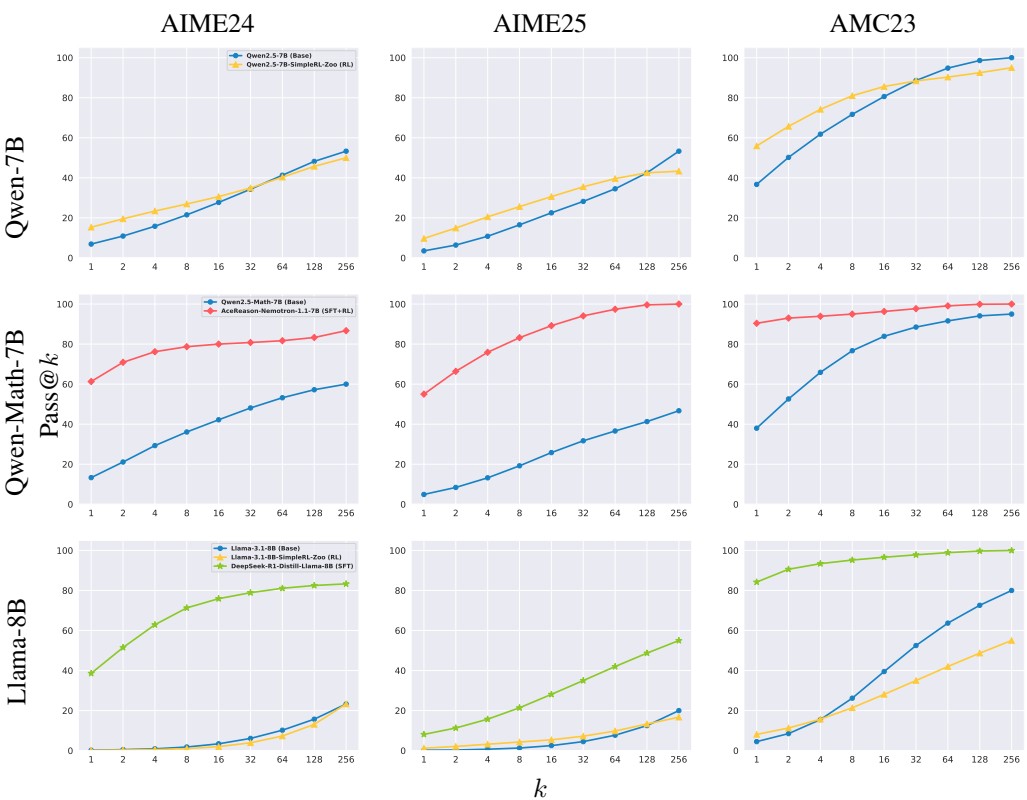

Figure 15: **Pass@$k$ performance curves for additional models.** Models are from Table 2 that are not included in Table 1 and datasets are AIME24, AIME25, and AMC23.

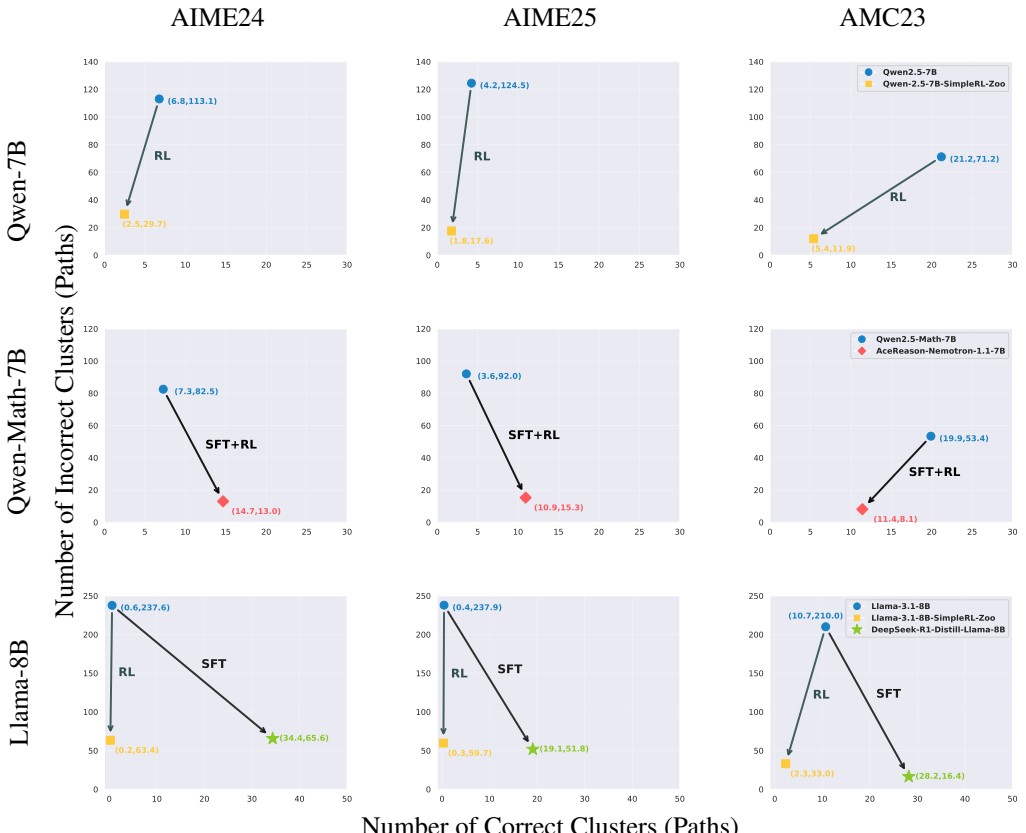

Figure 16: **Additional Result on the Effect of RL and SFT on the Number of Unique Paths.** The x-axis represents the number of correct clusters and the y-axis represents the number of incorrect clusters for trajectories before and after training of models in Table 2 that are not included in Table 1.

## C.5 DIFFERENT SIMILARITY METRIC AND THRESHOLD.

Using BLEU as the similarity matrix, we performed identical experiments at the trajectory-level. The results were consistent with those obtained using chrF and 60 as a threshold for hierarchical clustering. Furthermore, experiments with varying similarity thresholds also yielded identical results, as shown in Figure 17.

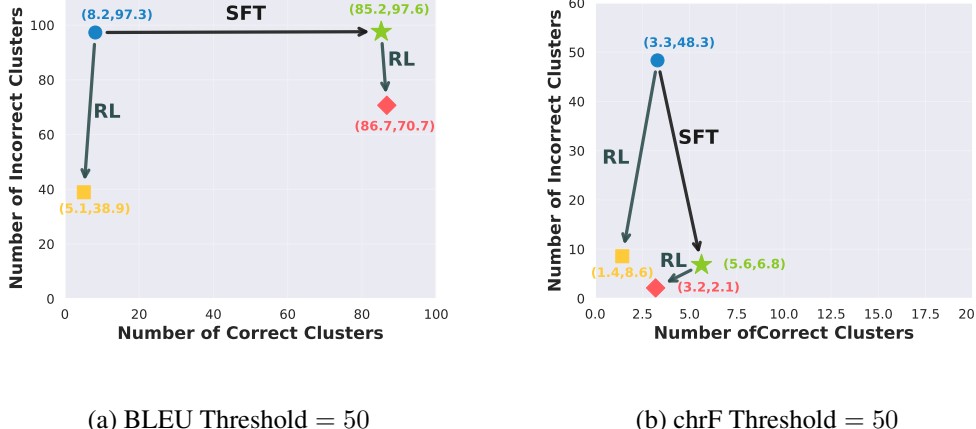

$$(a) \text{ BLEU Threshold} = 50 \qquad (b) \text{ chrF Threshold} = 50$$

Figure 17: **Number of Correct and Incorrect Clusters (Paths).** (a) Different similarity metric: BLEU and (b) Different Threshold of 7B models in Table 1 on AIME24.

## C.6 SIMILARITY DISTRIBUTION

Figure 18 illustrates the distributional characteristics of the upper triangular matrix elements (diagonal components excluded) derived from the similarity matrices in chrF-based clustering for unique path construction across models in Table 1. The results demonstrate that RL yields a notable increase in similarity measurements.

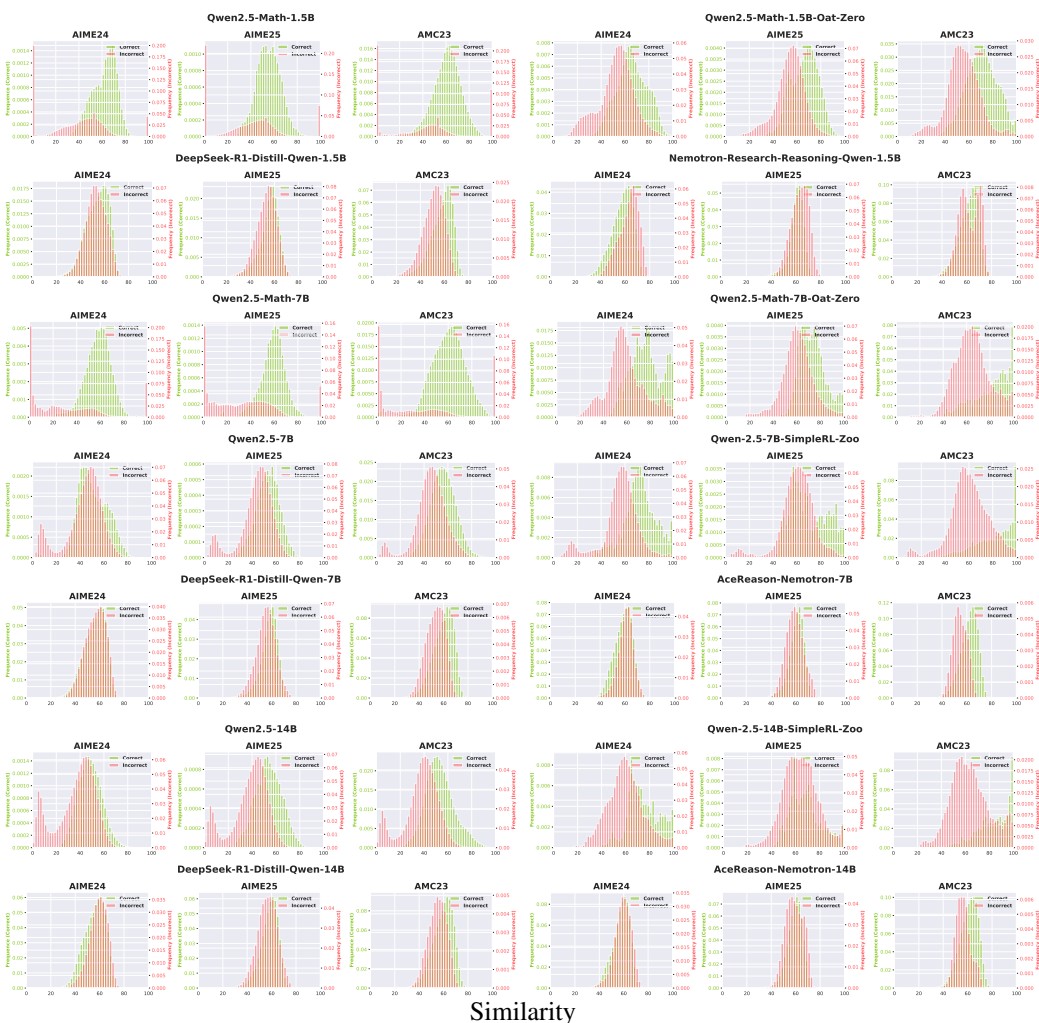

Figure 18: **Trajectory Similarity Distribution.** Frequency distribution of the upper triangular matrix obtained by extracting the diagonal elements from each model's similarity matrix across different models in Table 1 and three datasets, AIME24, AIME25, and AMC23.

## C.7 REPRESENTATIVES OF SFT

In this study, we treat the models distilled from DeepSeek-R1(Guo et al., 2025) as SFT models (Table 2). However, since these models generate multiple responses per problem during distillation, we conduct trajectory-level analysis for the case where SFT distillation uses a single response per problem. We performed SFT on the Qwen2.5-Math-1.5B in Table 1 using the s1k-1.1 dataset from Muennighoff et al. (2025). We used the Adam optimizer with a $10^{-5}$ learning rate, applying a cosine decay schedule and a weight decay of $10^{-4}$, and trained the model for 5 epochs with a maximum sequence length (block size) of 20000.

We generated M=256 responses for AIME24, AIME25, and AMC23, computed pairwise similarities with chrF, clustered them using UPGMA with 60 as a threshold, and calculated the number of unique trajectories. The results are shown in Figure 19. This figure shows that RL reduces the number of unique trajectories in incorrect outputs, while SFT increases the number of unique trajectories in correct outputs. This confirms that RL continues to compress incorrect trajectories, while SFT expands correct trajectories.

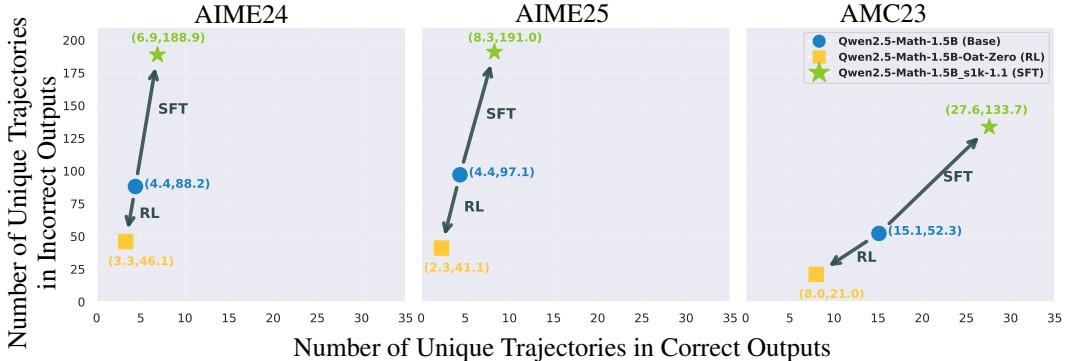

Figure 19: **Effect of RL and SFT on the Number of Unique Trajectories.** The x-axis represents the number of correct clusters and the y-axis represents the number of incorrect clusters for trajectories before and after training of 1.5B models.

## C.8 CODE DOMAIN

To validate our findings beyond the mathematical domain, we extend our experiments to code generation using HumanEval (Chen et al., 2021). For each problem, we generate $M = 128$ samples with 7B models in Table 1 , compute pairwise similarities using chrF, cluster them via UPGMA with a threshold of 70, and calculate the number of unique reasoning trajectories. We set sampling parameters to `temperature=0.6`, `top_p=0.95` and `max_tokens=16000`. We use Qwen Template for Qwen2.5-Math-7B and Qwen2.5-Math-Oat-Zero, and R1 Template for DeepSeek-R1-Distill-Qwen-7B and AceReason-Nemotron-7B. U+2581 and U+FF5C are replaced with '‥' and '—' in Figure 20.

---

**Qwen Template**

```
<|im_start|>system
You are a helpful assistant.<|im_end|>
<|im_start|>user
{input}
Please reason step by step, and complete the above Python
↪   function.<|im_end|>
<|im_start|>assistant
```

---

**R1 Template**

```
<|begin__of__sentence|><|User|>{input}
Please reason step by step, and complete the above Python
↪   function.<|Assistant|><think>
```

---

Figure 20: Prompt Templates for HumanEval.

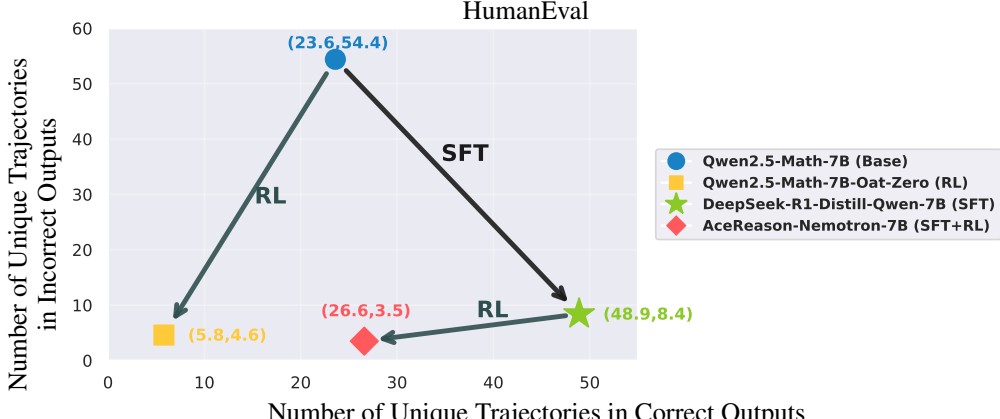

Figure 21: **Effect of RL and SFT on the Number of Unique Trajectories.** The x-axis represents the number of correct clusters and the y-axis represents the number of incorrect clusters for trajectories before and after training of 7B models in Table 1 on HumanEval.

These findings hold in the code domain as well, where we observe consistent trends: RL reduces the number of unique trajectories in incorrect outputs, while SFT increases the number of unique trajectories in correct outputs. In the code domain, RL compresses incorrect trajectories, while SFT expands correct trajectories.

# D  STEP-LEVEL ANALYSIS

## D.1  REASONING GRAPH CONSTRUCTION

**Graph Construction.**  Following Bogdan et al. (2025), we extract the trajectory up to the `</think>` token, then split it into sentences, using delimiters `.`, `?`, `!` (only when followed by a space) or `\n\n`, `\r\n\r\n`. If a chunk exceeds 300 tokens, forcibly split it, and if a chunk is under 10 tokens, merge it with the previous chunk. We used RAPIDS cuML's GPU-accelerated `KMeans` with scalable k-means++ initialization, running 10 restarts (`n_init = 10`) and capping each run at 300 iterations (`max_iter = 300`), with 2000 clusters (`n_clusters=2000`). The example for a node is in Table 3, and that for a graph is in Figure 22 and Figure 23.

| Node | Examples |
|---|---|
| #3 in 14B (Uncertainity Management) | Hmm, maybe this is getting too convoluted. Wait, perhaps this approach is also getting too messy. (DeepSeek-R1-Distill-Qwen-14B) Wait, perhaps it's getting too tangled. Wow, this is getting messy. (AceReason-Nemotron-14B) |
| #64 in 7B (Causal Aggregation) | Therefore, $y = \frac{-13650}{11} \div \left(-65\sqrt{14}\right) = \frac{13650}{11 \cdot 65\sqrt{14}}$. (DeepSeek-R1-Distill-Qwen-7B) Then, substitute $x = 4.5$ into equation 1: $84 \times \frac{9}{2} + 11\sqrt{14}\,y = 0 \Rightarrow 84 \times 4.5 + 11\sqrt{14}\,y = 0 \Rightarrow 84 \times 4.5 = 378$. (Qwen2.5-Math-7B-Oat-Zero) So plug $x = \frac{305t}{22}$, $y = \frac{15\sqrt{14}\,t}{11}$ into $x^2 + y^2 - 5x + Ey = 0$. (AceReason-Nemotron-7B) |
| #1384 in 7B (Interim Summary) | Now, we have equations of tangents at B and C: (AceReason-Nemotron-7B) So, if we let the tangents from $A$ to the points of tangency on $AB$ and $AC$ be $a$ and $b$ respectively, then $a + b = 5 + 10 - 9 = 6$. (Qwen2.5-Math-7B) So, to recap, tangent at $B$: $28x - 13\sqrt{14}y - 140 = 0$. (DeepSeek-R1-Distill-Qwen-7B) |

Table 3: **Representative Node Examples.** We converted all mathematical expressions to TEX.

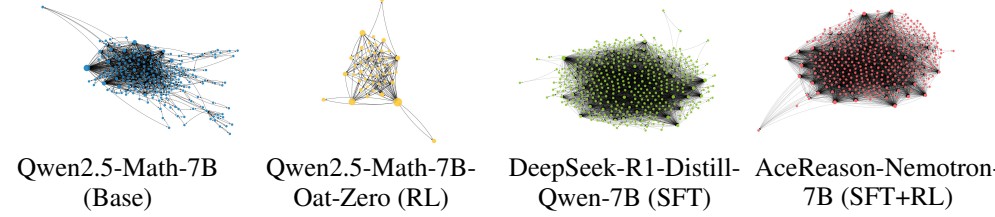

| Qwen2.5-Math-7B (Base) | Qwen2.5-Math-7B-Oat-Zero (RL) | DeepSeek-R1-Distill-Qwen-7B (SFT) | AceReason-Nemotron-7B (SFT+RL) |

Figure 22: **Visualization of Reasoning Graphs.** Results of 7B models in Table 1 on AIME24 Problem #1 Node encodes the node occurrence count, and edge length has no meaning. Graphs are visualized using Kamada-Kawai layout in NetworkX.

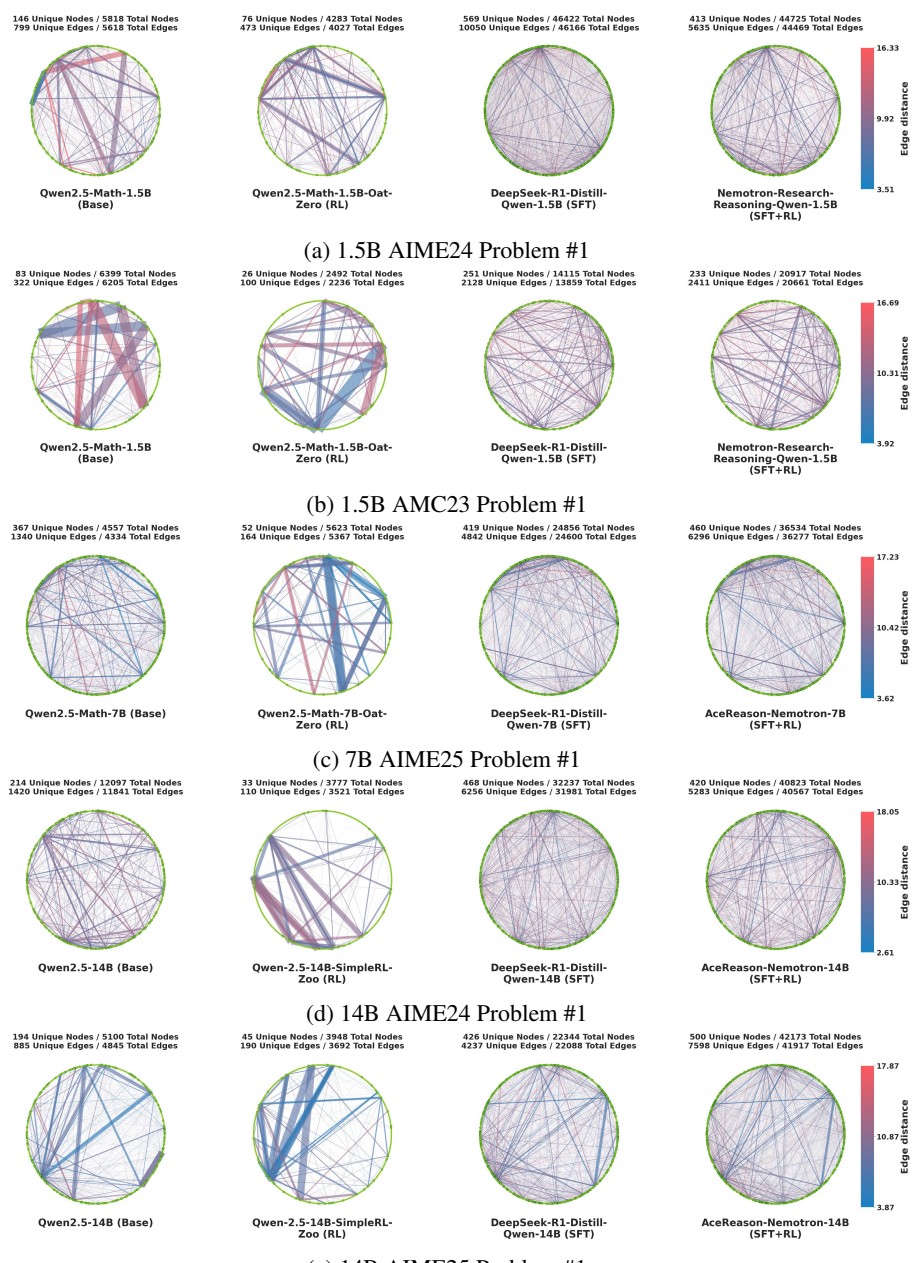

Figure 23: **Reasoning Graphs Examples.** Nodes arranged sequentially on circle, consistent across examples. Edge thickness encodes transition frequency, and edge color encodes edge distance.

## D.2 REASONING GRAPH ANALYSIS

**Estimation of the Exponential Decay Rate.** We illustrate in Figure 24 the rank plots obtained for *Visitation Frequency*, *Degree*, and *Betweenness Centrality*. We show in Figure 25 an illustrative example of linear regression analysis performed on the log-linear plot to estimate the exponential decay rate $\beta$.

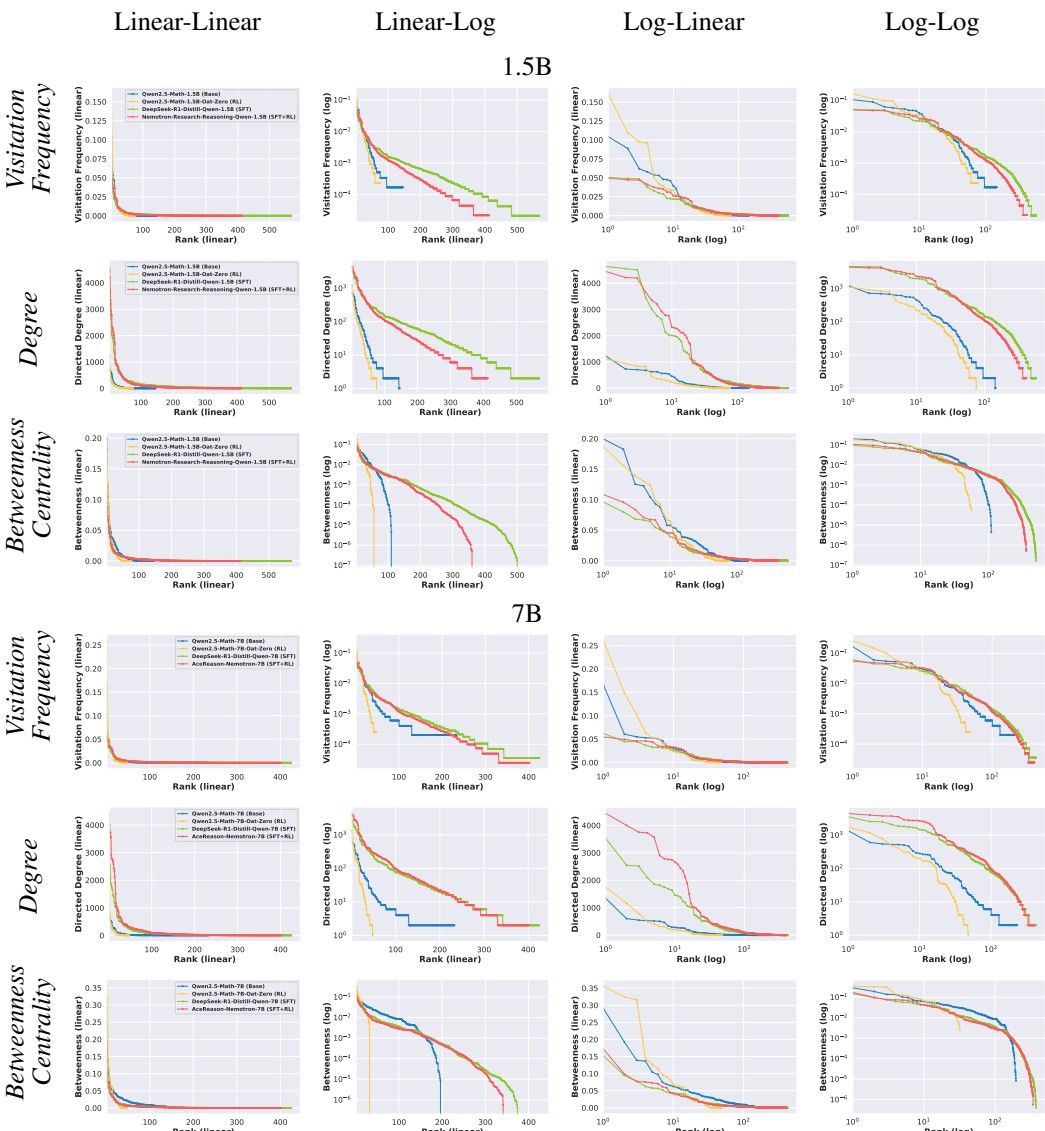

Figure 24: **Rank plots for *Visitation Frequency*, *Degree*, and *Betweenness Centrality*.** Results for 1.5B (top) and 7B (bottom) models in Table 1 on AIME24 Problem #1.

**Complete Results for $\beta$.** The box plots of estimated exponential decay rates $\beta$ for *visitation frequency*, *degree*, and *betweenness centrality* across Base, RL, SFT, and SFT+RL models on AIME24, AIME25, and AMC23 datasets from Table 1 are shown in Figure 26. The results demonstrate that RL training substantially increases all metrics relative to the Base model, while SFT reduces them.

**Edge Distance Distribution.** The differences in edge distance ($L_2$ norm of the centroid of sentence vectors for each node) across Base, RL, SFT, and SFT+RL models are shown in Figure 27. However, No clear differences in edge distance distribution were observed.

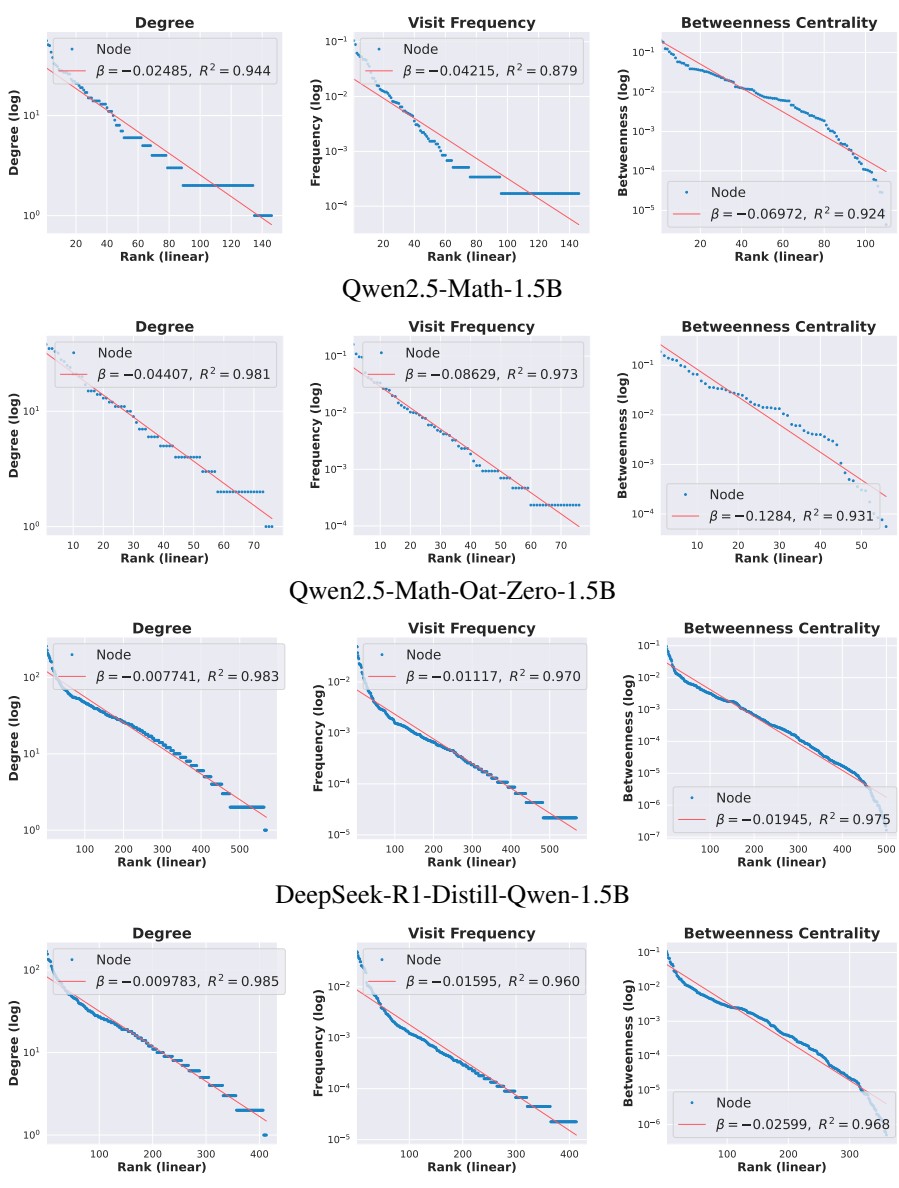

Figure 25: **Example of Exponential Decay Rate Estimation.** Linear regression on a semi-log plot of *degree*, *visitation frequency*, and *betweenness centrality* in AIME24 Problem #1.

**Inter-model Similarity in Node Visitation Frequency.** To investigate how RL and SFT modify the reasoning graph, Figure 28 presents scatter plots of node *visitation frequencies* between pairs of models. Points closer to the line $y = x$ indicate that the two models utilize nodes with similar visitation frequencies.

We employ the symmetric Mean Absolute Percentage Error (sMAPE) as a quantitative measure:

$$\text{sMAPE} = \frac{100}{n} \sum_{t=1}^{n} \frac{|y_t - x_t|}{(|y_t| + |x_t|)/2}$$

where $n$ represents the total number of nodes, $x_t$ denotes the visitation frequency of node $t$ ($t = 1, \ldots, n$) for the model on the x-axis, and $y_t$ represents the visitation frequency of node $t$ for the model on the y-axis.

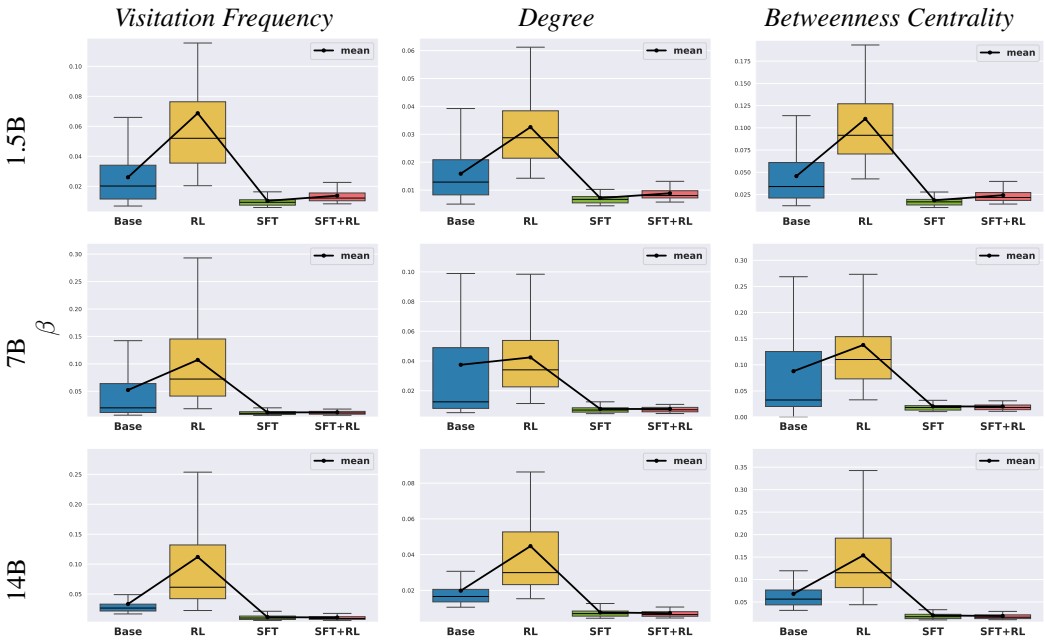

Figure 26: **Exponential Decay Rate for Visitation Frequency, Degree, Betweenness Centrality.**
Box plots show the estimated exponential decay rate $\beta$ across all problems separated by 1.5B models
in Table 1 and datasets, AIME24, AIME25, AMC23.

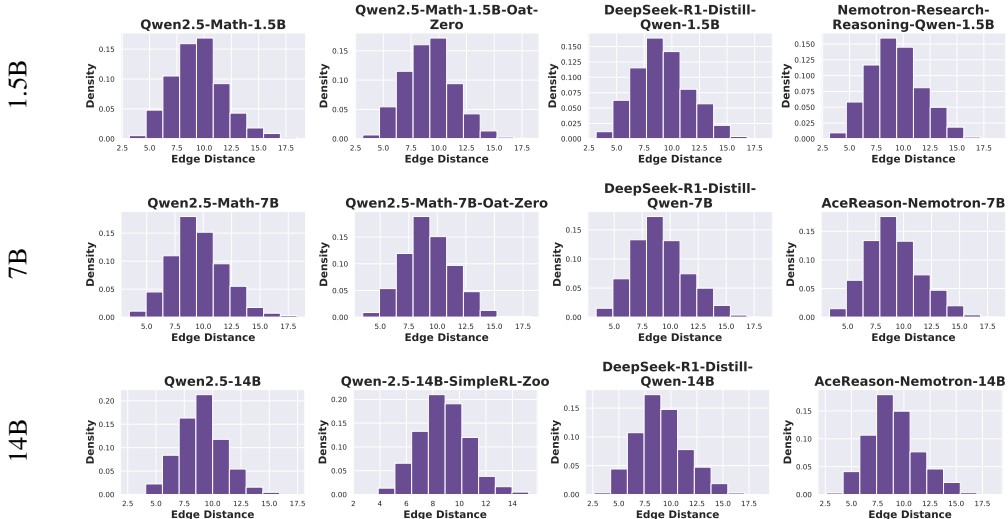

Figure 27: **Edge Distance Distribution.** Frequency distribution of edge distances shown in bins
across models in Table 1 and datasets, AIME24, AIME25, AMC23.

This reveals distinct behavioral patterns: Base vs. RL and SFT vs. SFT+RL exhibit relatively low
sMAPE values, indicating that RL does not substantially alter the set of visited nodes compared to
the pre-RL models. In contrast, Base vs. SFT demonstrates a considerably higher sMAPE, suggest-
ing that SFT significantly modifies the node visitation patterns relative to the base model.

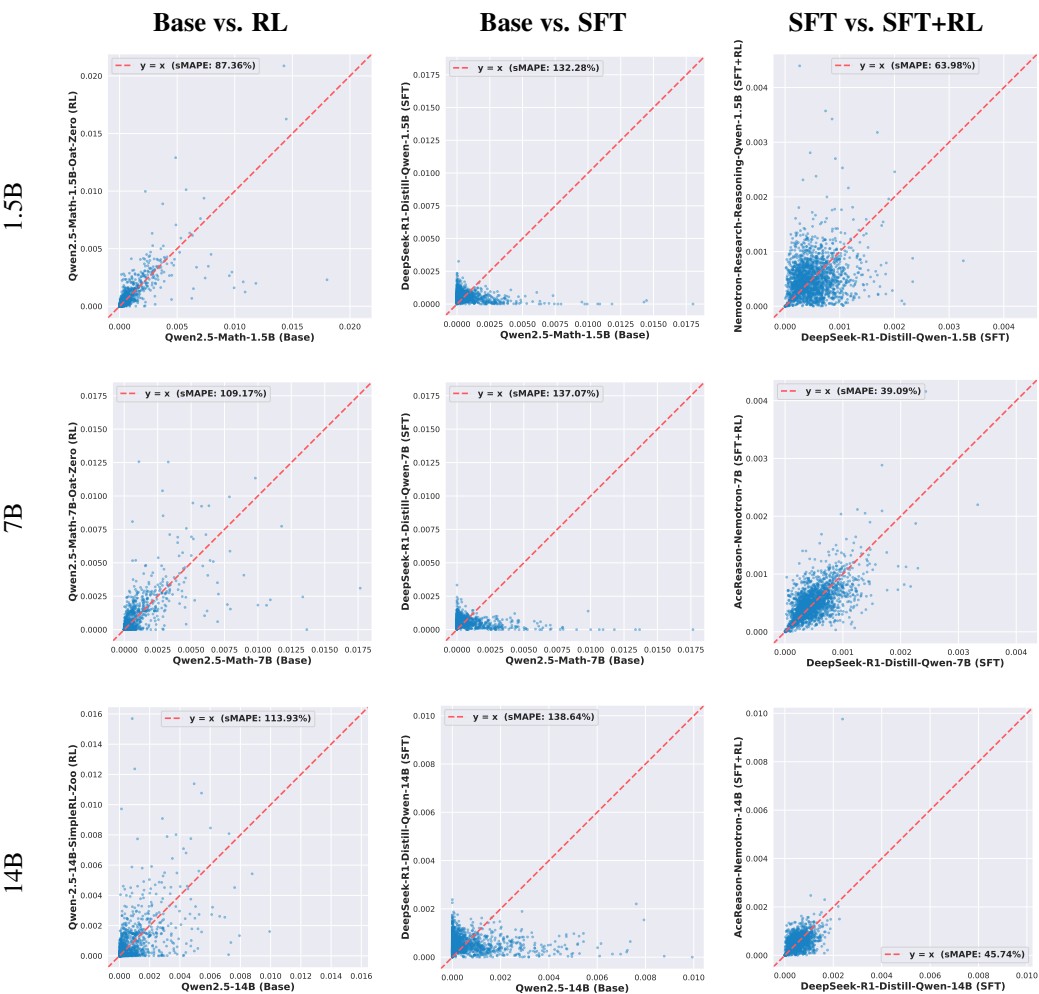

Figure 28: **Visitation Frequency Scatter Plot Between Two Models.** Each plot represents a node, with coordinates indicating the visitation frequency in the integrated graph across all AIME24 problems for two models in Table 1. The closer the plots are to the line $y = x$, the more similar ther visitation frequency of that node between two models.

### D.3 STRUCTURAL GRAPH PROPERTIES

**Graphlet Analysis.** We utilize graphlets (Milo et al., 2004; Pržulj et al., 2004) to analyze the local structure of graphs. Graphlets have been extensively applied across diverse domains, including protein interaction networks (Pržulj et al., 2006; Pržulj, 2007), social network (Janssen et al., 2012), and world trade networks (Sarajlić et al., 2016). Since counting 5-node graphlets is computationally hard and 3-node graphlets consist of only two types, which is insufficient to describe graph structures, we focus on 4-node graphlets and count the subgraphs shown in Figure 9a in the reasoning graph integrated across all problems $\mathcal{G}^l$ for each dataset. We then calculate the proportions of 4-node graphlets and compare them across models. As shown in Figure 29, across all models and datasets, RL consistently decreases linear graphlets G3 and G4 while increasing cyclic G7 and G8. G5 also shows a slight increase. SFT that imitates DeepSeek-R1's reasoning traces exhibits similar increases in G7 and G8, suggesting that the teacher model's reasoning graph possesses comparable structural tendencies. Notably, all models except base models show highly similar graphlet proportions. However, their accuracies diverge significantly (see Figure 12). This indicates that despite similar local reasoning graph structures between RL-trained models from Base and SFT or SFT+RL models, substantial performance gaps persist in reasoning capabilities.

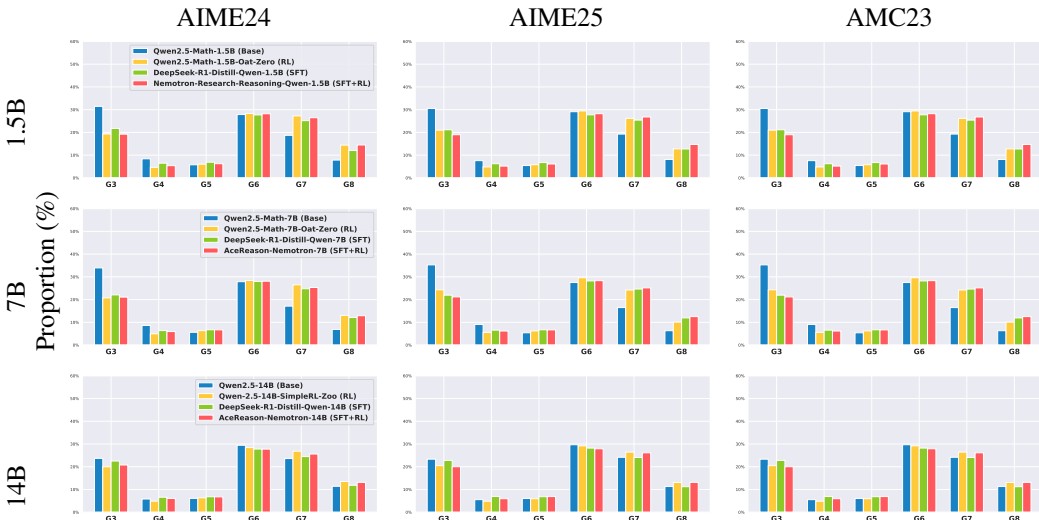

Figure 29: **Poroportion of 4-node Graphlets.** Graphlet proportions for models in Table 1 on AIME24, AIME25, and AMC23.

**Global Structures of Reasoning Graph.** We calculated edge density, clustering coefficient (Watts & Strogatz, 1998), assortativity (Newman, 2002), modularity (Girvan & Newman, 2002), Freeman centralization (Freeman, 1978), average path length (Watts & Strogatz, 1998), global efficiency (Latora & Marchiori, 2001), and algebraic connectivity (Fiedler, 1973) for each model and each problem to examine differences in complex network structure (Newman, 2003), and averaged these metrics within the dataset. We utilized NetworkX library. The results are shown in Figure 31.

Here, we consider an undirected graph $\mathcal{G} = (\mathcal{V}, \mathcal{E})$.

Edge density is given by

$$\rho(\mathcal{G}) = \frac{2|\mathcal{E}|}{|\mathcal{V}|(|\mathcal{V}|-1)}.$$

Edge density is ratio of the number of observed edges to the maximum possible number of edges in the graph.

The local clustering coefficient (Watts & Strogatz, 1998) is given by

$$C_i(\mathcal{G}) = \frac{t_i}{k_i(k_i - 1)},$$

where $t_i$ denotes the number of triangles and $k_i$ is degree involving node $i$. The global clustering coefficient is given by

$$C(\mathcal{G}) = \frac{1}{|\mathcal{V}|} \sum_{i \in \mathcal{V}} C_i.$$

Clustering coefficient is the proportion of observed connections among the neighbors of a node relative to the number of possible connections over random the graph. To compare graphs of different sizes, we normalize $C(\mathcal{G})$ by the average length $C(\mathcal{G}_{\text{rand}})$ of a random graph.

Assortativity (Newman, 2002) is given by

$$R(\mathcal{G}) = \frac{\sum_{i,j \in \mathcal{V}} \left( A_{i,j} - \frac{k_i k_j}{2|\mathcal{E}|} \right) k_i k_j}{\sum_{i,j \in \mathcal{V}} \left( k_i \delta_{i,j} - \frac{k_i k_j}{2|\mathcal{E}|} \right) k_i k_j},$$

where $A_{i,j} = 1$ if there is an edge between $i$ and $j$, $k_i = \sum_j A_{i,j}$ is the degree of node $i$. Assortativity is the Pearson correlation coefficient between the degrees of nodes at the ends of edges. A highly assortative network is one where high-degree nodes connect with other high-degree nodes, and low-degree nodes connect with other low-degree nodes. In contrast, a disassortative network has a hub structure, where high-degree nodes are connected to low-degree nodes.

For a partition $\{c_i\}$, modularity (Girvan & Newman, 2002) is given by

$$Q(\mathcal{G}) = \frac{1}{2|\mathcal{E}|} \sum_{i,j \in \mathcal{V}} \left( A_{i,j} - \frac{k_i k_j}{2|\mathcal{E}|} \right) \delta(c_i, c_j),$$

where $A_{i,j} = 1$ if there is an edge between $i$ and $j$, $k_i = \sum_j A_{i,j}$ is the degree of node $i$, and $\delta(c_i, c_j) = 1$ if $i$ and $j$ belong to the same community. Modularity measures the strength of division of a network into communities, relative to a random graph.

Freeman centralization (Freeman, 1978) is given by

$$C_D(\mathcal{G}) = \frac{\sum_{i \in \mathcal{V}} (d_{\max} - d(i))}{(|\mathcal{V}|-1)(|\mathcal{V}|-2)},$$

where we use the denominator $(|\mathcal{V}|-1)(|\mathcal{V}|-2)$ to normalize to ranges between 0 and 1. $(|\mathcal{V}|-1)(|\mathcal{V}|-2)$ corresponds to the value achieved by a star graph. Freeman centralization quantifies the extent to which the network's connectivity is organized around a central node.

Average path length (Watts & Strogatz, 1998) is given by

$$L(\mathcal{G}) = \frac{1}{|\mathcal{V}|(|\mathcal{V}|-1)} \sum_{i,j \in \mathcal{V}, i \neq j} d(i,j),$$

where $d(i,j)$ denotes the length of the shortest path between nodes $i$ and $j$. Average path length is the mean of the shortest path lengths between all pairs of nodes in the network. To compare graphs of different sizes, we normalize $L(\mathcal{G})$ by the average length $L(\mathcal{G}_{\text{rand}})$ of a random graph.

Global efficiency (Latora & Marchiori, 2001) is given by

$$E(\mathcal{G}) = \frac{1}{|\mathcal{V}|(|\mathcal{V}|-1)} \sum_{i,j \in V, i \neq j} \frac{1}{d(i,j)},$$

where $d(i,j)$ denotes the length of the shortest path between nodes $i$ and $j$ and we use $(\mathcal{V}-1)(\mathcal{V}-2)$ for normalization. Global efficiency is the mean of the inverse shortest path length across all node pairs, indicating communication efficiency.

Algebraic connectivity is given by the second smallest eigenvalue of the graph Laplacian, which reflects the robustness of network connectivity. Low algebraic connectivity indicates that the graph can be easily disconnected into separate components by removing only a few edges or vertices.

The small-world index (Watts & Strogatz, 1998) can be obtained as $\sigma(\mathcal{G}) = \frac{C(\mathcal{G})/C(\mathcal{G}_{\text{rand}})}{L(\mathcal{G})/L(\mathcal{G}_{\text{rand}})}$. Minegishi et al. (2025) analyzed the small-world index of reasoning graphs.

**Relationship with Pass@$k$.** Comparing the results in Figure 30 and Figure 31, we observe that Pass@1 / Pass@$k$ is positively correlated with *Global Efficiency* and *Algebraic Connectivity*, while negatively correlated with *Modularity*. A higher Pass@1 / Pass@$k$ ratio indicates that the improvement from Best-of-$k$ sampling over single inference is marginal, suggesting that the model can effectively explore the solution space and reach the correct answer in a single attempt. This reasoning capability is associated with graph structures that exhibit low modularity and facilitate traversal across the entire graph, enabling efficient navigation between distant nodes.

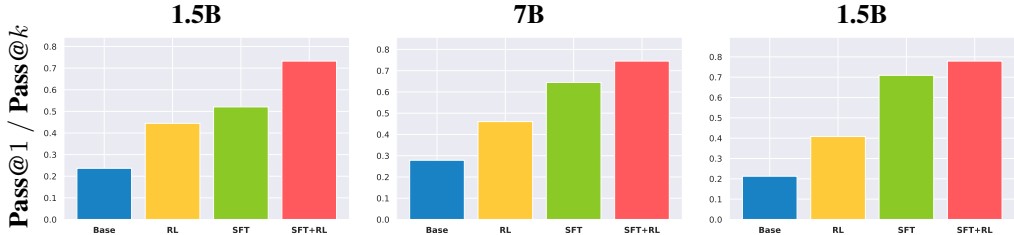

Figure 30: **Pass@1 / Pass@$k$ performance by model.** The average Pass@1 / Pass@$k$ across AIME24, AIME25, and AMC23 for each model in Table 1.

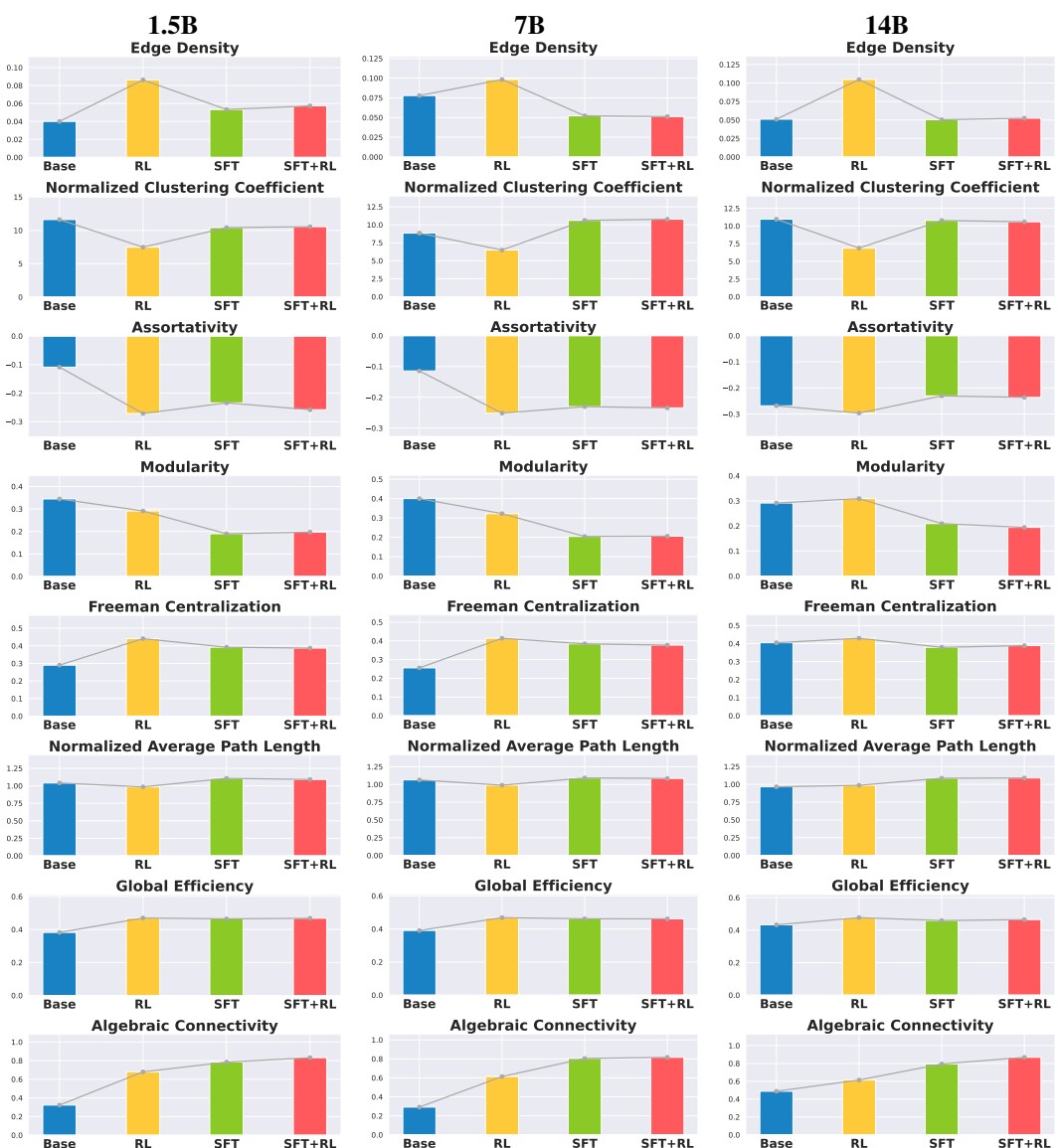

Figure 31: **Graph Structural Metrics of Reasoning Graphs.** Each model size in Table 1 shows mean values (averaged across AIME24, AIME25, AMC23) for eight core graph structural metrics: edge density, clustering coefficient, assortativity, modularity, Freeman centralization, average path length, global efficiency, and algebraic connectivity.

## D.4 ABLATION OF REASONING GRAPH

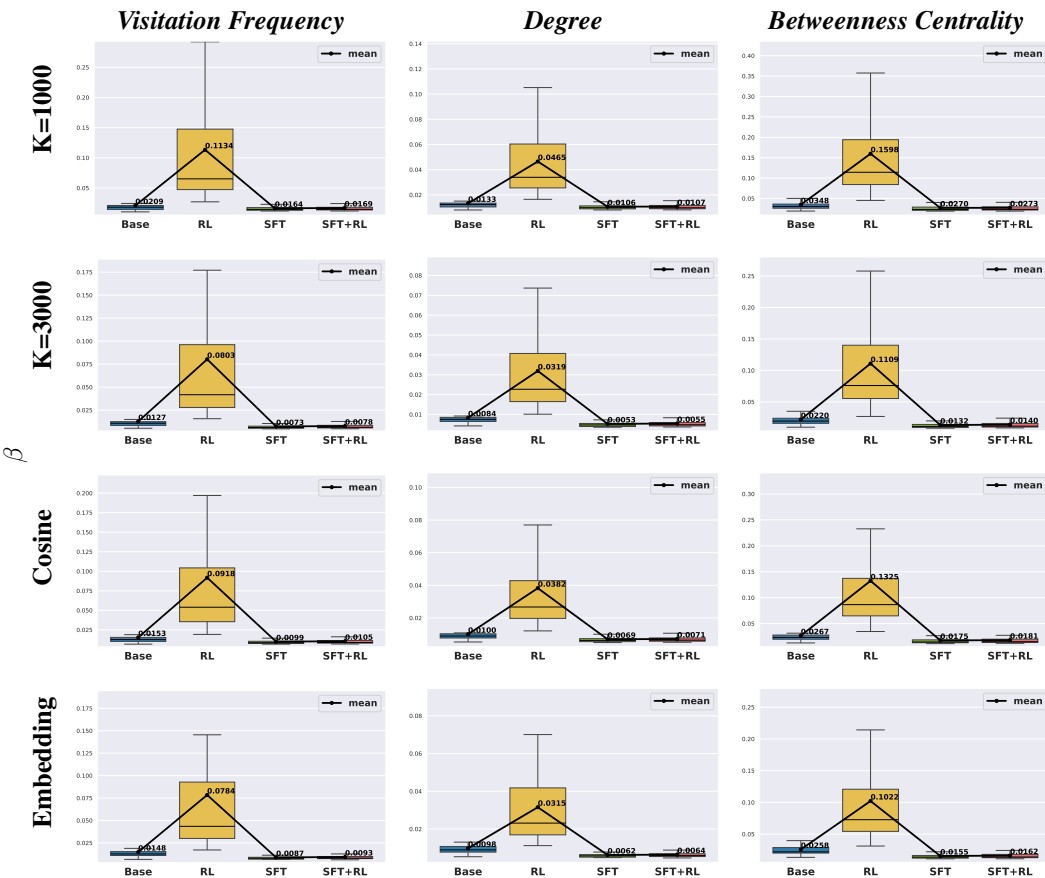

Figure 32: **Exponential Decay Rate for *Visitation Frequency*, *Degree*, *Betweenness Centrality*.** Box plots show the estimated exponential decay rate $\beta$ across all problems in AIME24 for the 7B models in Table 1. K=1000 and K=3000 denote the number of clusters in $K$-means clustering, 'cosine' indicates the use of cosine distance in $K$-means, and 'embedding' refers to the use of GTE-base-en-v1.5 for sentence embeddings.

We conduct an ablation study to assess the impact of different reasoning graph construction methods on our results. In the main results, we construct reasoning graphs by first segmenting generated reasoning traces into individual sentences, obtaining sentence embeddings using BGE-large-en-v1.5 (Xiao et al., 2024), and clustering them via $K$-means with $K = 2000$ and L2 norm to define graph nodes. We systematically vary the number of clusters $K$, the distance metric, and the sentence embedding model to analyze how these design choices affect the exponential decay rates underlying the "RL squeezes, SFT expands" phenomenon. We estimate the exponential decay rate $\beta$ for the 7B models on AIME24 (Table 1) under three alternative configurations: (i) varying the cluster count to $K = 1000$ and $K = 3000$, (ii) replacing the distance metric with cosine distance, computed via L2 normalization of sentence embeddings, and (iii) substituting the embedding model with GTE-base-en-v1.5 (Zhang et al., 2024), which produces $d = 768$ dimensional representations.

Figure 32 shows that across all ablations, RL consistently increases the mean exponential decay rate $\beta$, while SFT decreases it. This pattern remains consistent with the main results presented in Figure 26.

## D.5 SPARSIFYING REASONING GRAPHS

As described in Section 4.1, we defined nodes by clustering sentence embeddings and analyzed the properties of reasoning graphs. Figure 31 shows that the edge density is approximately 0.1 for the RL model and 0.05 for others, indicating these are not sparse graphs. Nevertheless, we examine the behavior of these metrics as graph sparsity increases. We construct reasoning graphs for the 7B models in Table 1 following Section 4.1 and apply distance-based sparsification in the sentence embedding space. We sort the edges connected to each node by L2 norm and retain only the top-10 or top-20 closest edges. Nodes with a degree under 10 or 20 retain all their edges.

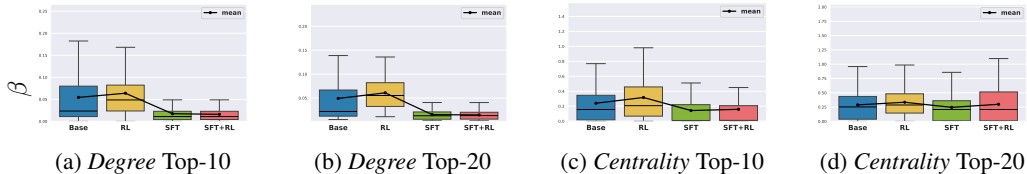

(a) *Degree* Top-10    (b) *Degree* Top-20    (c) *Centrality* Top-10    (d) *Centrality* Top-20

Figure 33: **Exponential Decay Rates of *Degree* and *Betweenness Centrality (Centrality)* on Sparsified Reasoning Graphs.** The box plots aggregate the estimated decay rate $\beta$ for the 7B models in Table 1, after sparsifying the graphs using the top-10 and top-20 distance–based sparcification. Results are combined across AIME24, AIME25, and AMC23.

Figure 33 presents box plots of the exponential decay rates $\beta$, estimated across AIME24, AIME25, and AMC23 for the 7B models in Table 1. Figure 33 shows that when sparsifying based on sentence embedding distance (top-10 and top-20), RL increases the exponential decay rate $\beta$ while SFT decreases it. These results are consistent with the non-sparsified results in Figure 6. An important caveat is that our graph construction method (Section 4.1) ensures the reasoning graph is weakly connected by designating problem $x$ as the initial node. However, sparsification may disconnect the graph and create unreachable nodes whose betweenness centrality becomes zero.

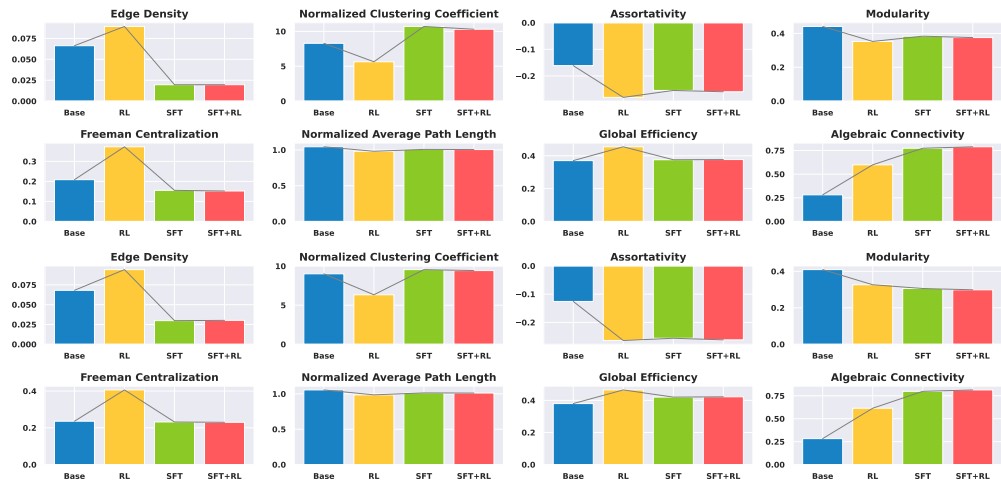

Figure 34: **Comparison of Eight Graph Metrics for the Sparcified Graphs with the Top-10 (Up) and Top-20 (Bottom) Distance-based Sparcification Method across Base, RL, SFT, and SFT+RL Models (7B).** Values are averaged across three datasets, AIME24, AIME25, and AMC23.

## D.6 Representatives of SFT

Following Appendix C.7, we conduct step-level analysis for RL and SFT when performing SFT with one response per problem on s1k-1.1 dataset(Muennighoff et al., 2025)

After constructing the reasoning graphs following Section 4.1 and Appendix D.1 with $M = 256$ and $K = 1000$, we estimated the exponential decay rates of *Visitation Frequency*, *Degree*, and *Betweenness Centrality* for Qwen2.5-Math-1.5B (Base), Qwen2.5-Math-1.5B-Oat-Zero (RL), and Qwen2.5-Math-1.5B-s1k-1.1 (SFT) on AIME24, AIME25, and AMC23. Consistent with Figure 6, RL exhibits higher decay rates, whereas SFT exhibits lower decay rates.

| *Visitation Frequency* | *Degree* | *Betweenness Centrality* |

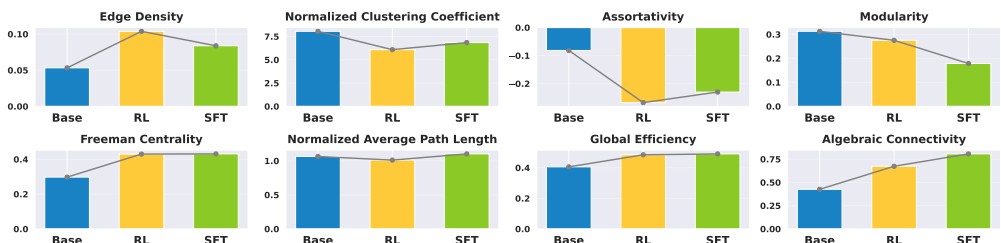

Figure 35: **Exponential Decay Rate for *Visitation Frequency*, *Degree*, *Betweenness Centrality*.** Box plots show the estimated exponential decay rate $\beta$ across all problems in AIME24, AIME25, and AMC23 for the 1.5B models in Appendix D.6

Figure 36: **Comparison of Eight Graph Metrics across Base, RL, and SFT models (1.5B).** Values are averaged across three datasets, AIME24, AIME25, and AMC23.

We then computed the topological metrics of the reasoning graphs and present them in Figure 36. The results exhibit trends consistent with Figure 8.

## D.7 CODE DOMAIN

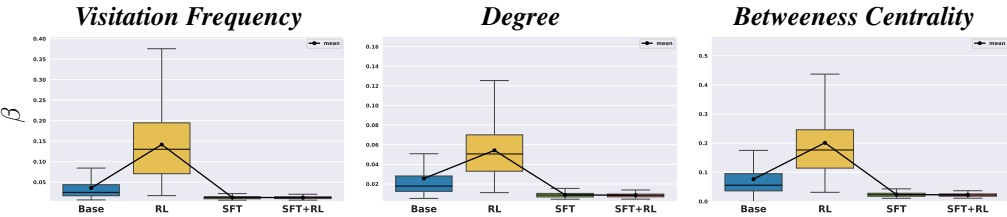

Figure 37: **Exponential Decay Rate for *Visitation Frequency*, *Degree*, *Betweenness Centrality***
Box plots show the estimated exponential decay rate $\beta$ across all problems in HumanEval on the
7B models in Table 1.

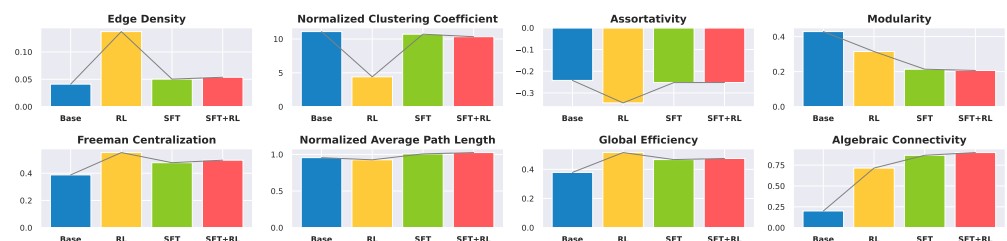

Figure 38: **Comparison of Eight Graph Metrics across Base, RL, and SFT models (7B) on
HumanEval.**

In addition to Appendix C.8, to assess the generalizability of our findings across domains, we conduct experiments on code generation using HumanEval (Chen et al., 2021). For each problem instance, we generate $M = 128$ samples with the 7B models in Table 1. We set sampling parameters to `temperature=0.6`, `top_p=0.95` and `max_tokens=16000`. We use Qwen Template for Qwen2.5-Math-7B and Qwen2.5-Math-Oat-Zero, and R1 Template for DeepSeek-R1-Distill-Qwen-7B and AceReason-Nemotron-7B. Then, we construct reasoning graphs following the same methodology described in Section 4.1.

Figure 37 shows box plots of the exponential decay rates of the rank plots of the reasoning graphs for each HumanEval problem. We observe that, for visitation frequency, degree, and betweenness centrality, the exponential decay rate $\beta$ increases with RL, whereas $\beta$ decreases with SFT.

Furthermore, we compute eight topology metrics of the reasoning graph for each sample and report their averages over all samples in Figure 38 In this domain, we obtained results consistent with those in Figure 8.

## E    EXAMPLES OF TRAJECTORY-LEVEL ANALYSIS

An example where incorrect trajectories from Qwen2.5-Math-7B-Oat-Zero (RL) are similar to each other, while those from DeepSeek-R1-Distill-Qwen-7B (SFT) are dissimilar.

---

**Qwen2.5-Math-7B-Oat-Zero (RL)   AIME24 Problem #2**
**m=182 (vs. 230) chrF=91.81 (Incorrect)**

To solve this problem, we need to use some geometric properties and algebra. Here's the step-by-step solution:
1. **Identify the Power of a Point:** The power of point $A$ with respect to the circle $\omega$ can be used to find the length $AP$. The power of point $A$ is given by $AB^2 = AP \cdot AD$.
2. **Find the Length $AD$:** Since $D$ is the intersection of the tangents to $\omega$ at $B$ and $C$, $D$ is the symmedian point of triangle $ABC$. The length $AD$ can be found using the formula for the symmedian point:

$$AD = \frac{2bc}{b^2 + c^2} \cdot \sqrt{bc\left(1 - \frac{a^2}{(b+c)^2}\right)}$$

where $a = BC = 9$, $b = AC = 10$, and $c = AB = 5$.
3. **Calculate $AD$:**

$$AD = \frac{2 \cdot 10 \cdot 5}{10^2 + 5^2} \cdot \sqrt{10 \cdot 5\left(1 - \frac{9^2}{(10+5)^2}\right)} = \frac{100}{125} \cdot \sqrt{50\left(1 - \frac{81}{225}\right)} = \frac{4}{5} \cdot \sqrt{50 \cdot \frac{144}{225}}$$

$$= \frac{4}{5} \cdot \sqrt{\frac{50 \cdot 144}{225}} = \frac{4}{5} \cdot \sqrt{\frac{7200}{225}} = \frac{4}{5} \cdot \sqrt{32} = \frac{4}{5} \cdot 4\sqrt{2} = \frac{16\sqrt{2}}{5}$$

4. **Find $AP$:** Using the power of point $A$:

$$AB^2 = AP \cdot AD \implies 5^2 = AP \cdot \frac{16\sqrt{2}}{5} \implies 25 = AP \cdot \frac{16\sqrt{2}}{5}$$

$$\implies AP = \frac{25 \cdot 5}{16\sqrt{2}} = \frac{125}{16\sqrt{2}} = \frac{125\sqrt{2}}{32}$$

To rationalize the denominator, we multiply the numerator and the denominator by $\sqrt{2}$:

$$AP = \frac{125\sqrt{2}}{32} = \frac{125 \cdot 2}{32 \cdot \sqrt{2} \cdot \sqrt{2}} = \frac{250}{64} = \frac{125}{32}$$

So, $AP = \frac{125}{32}$.
5. **Find $m$ and $n$:** Here, $m = 125$ and $n = 32$, and they are relatively prime. Therefore, $m + n = 125 + 32 = 157$.
The final answer is $\boxed{157}$.

---

**Qwen2.5-Math-7B-Oat-Zero (RL)    AIME24 Problem #2**
**m=(182 vs.) 230 chrF=91.81 (Incorrect)**

To solve this problem, we need to use some geometric properties and algebra. Here's the step-by-step solution:
1. **Identify the Power of a Point**: The power of point $A$ with respect to the circle $\omega$ can be used to find the length $AP$. The power of point $A$ is given by $AB^2 = AP \cdot AD$.
2. **Find the Length $AD$**: Since $D$ is the intersection of the tangents to $\omega$ at $B$ and $C$, $D$ is the symmedian point of $\triangle ABC$ with respect to $A$. The length $AD$ can be found using the formula for the symmedian length:

$$AD = \frac{2bc}{b^2 + c^2} \cdot \sqrt{bc\left(1 - \frac{a^2}{(b+c)^2}\right)}$$

where $a = BC = 9$, $b = AC = 10$, and $c = AB = 5$.

3. **Calculate $AD$**:

$$AD = \frac{2 \cdot 10 \cdot 5}{10^2 + 5^2} \cdot \sqrt{10 \cdot 5 \left(1 - \frac{9^2}{(10+5)^2}\right)} = \frac{100}{125} \cdot \sqrt{50 \left(1 - \frac{81}{225}\right)} = \frac{4}{5} \cdot \sqrt{50 \cdot \frac{144}{225}}$$

$$= \frac{4}{5} \cdot \sqrt{\frac{50 \cdot 144}{225}} = \frac{4}{5} \cdot \sqrt{\frac{7200}{225}} = \frac{4}{5} \cdot \sqrt{32} = \frac{4}{5} \cdot 4\sqrt{2} = \frac{16\sqrt{2}}{5}$$

4. **Find $AP$**: Using the power of point $A$:

$$AB^2 = AP \cdot AD \implies 5^2 = AP \cdot \frac{16\sqrt{2}}{5} \implies 25 = AP \cdot \frac{16\sqrt{2}}{5}$$

$$\implies AP = \frac{25 \cdot 5}{16\sqrt{2}} = \frac{125}{16\sqrt{2}} = \frac{125\sqrt{2}}{32}$$

5. **Rationalize the Denominator**: To express $AP$ in the form $\frac{m}{n}$, we rationalize the denominator:

$$AP = \frac{125\sqrt{2}}{32} = \frac{125 \cdot 2}{32 \cdot \sqrt{2} \cdot \sqrt{2}} = \frac{250}{64} = \frac{125}{32}$$

So, $AP = \frac{125}{32}$. The integers $m$ and $n$ are 125 and 32, respectively, and they are relatively prime. Therefore, $m + n = 125 + 32 = 157$.

The final answer is $\boxed{157}$.

---

**DeepSeek-R1-Distill-Qwen-7B (SFT)    AIME24 Problem #2**
**m=172 (vs. 196) chrF=29.56 (Incorrect)**

Okay, so I'm trying to solve this geometry problem. Let me read it again:

We have triangle $ABC$ inscribed in a circle $\omega$. The tangents to $\omega$ at $B$ and $C$ intersect at point $D$. Then, $\overline{AD}$ intersects $\omega$ again at $P$. We're given $AB = 5$, $BC = 9$, and $AC = 10$. We need to find $AP$ in the form $\frac{m}{n}$ where $m$ and $n$ are coprime, and then compute $m + n$.

Alright, let's break this down step by step. First, let me visualize the problem. We have triangle $ABC$ with sides $AB = 5$, $BC = 9$, and $AC = 10$. It's inscribed in a circle $\omega$, which means it's a cyclic quadrilateral if we consider the circle. The tangents at $B$ and $C$ meet at $D$, so $D$ is the ex-pole or something? Hmm, not sure. Maybe I need to recall some properties.

Since $D$ is the intersection of the tangents at $B$ and $C$, $D$ must lie on the polar of the line $BC$ with respect to circle $\omega$. Wait, maybe I should think in terms of power of a point.

Yes, the power of point $D$ with respect to $\omega$ should be equal to $DB^2 = DC^2$ since $DB$ and $DC$ are both tangent lengths from $D$ to $\omega$. So, $DB = DC$. That might be useful later.

Now, $AD$ intersects $\omega$ again at $P$. So, $AD$ is a secant of the circle, starting at $A$ and going through $P$. We need to find the length $AP$.

Given that, perhaps I can use the Power of a Point theorem. The Power of Point $A$ with respect to circle $\omega$ is $AB \times AC$ because $A$ is on the circle, so its power is zero. Hmm, that doesn't seem immediately helpful.

Wait, actually, $A$ is on the circle, so any line through $A$ intersecting the circle again will have power zero. So, the power of $A$ is zero, which means $AP \times AD = 0$. But that's trivial because $AD$ passes through $A$, so $AP \times AD$ is just $AP \times (AP + PD)$, but since $A$ is on the circle, $AP$ is the length from $A$ to $P$, and $PD$ is from $P$ to $D$. Hmm, maybe I need a different approach.

Alternatively, maybe I can use harmonic division or projective geometry concepts, but I'm not sure. Maybe coordinate geometry? That might be messy, but perhaps manageable.

Alternatively, maybe using inversion? Since we have tangents and circles, inversion could be useful. But that might be a bit advanced for me right now.

Wait, let's think about triangle $ABC$. We know all three sides: 5, 9, 10. Maybe we can compute some angles or the circumradius.

Yes, that might be a good starting point. Let me compute the circumradius $R$ of triangle $ABC$. Using the formula:

$R = \frac{abc}{4\Delta}$

Where $a, b, c$ are the sides, and $\Delta$ is the area of the triangle.

First, let's compute the area $\Delta$ using Heron's formula.

The semi-perimeter $s = \frac{5+9+10}{2} = \frac{24}{2} = 12$.

Then, $\Delta = \sqrt{s(s-a)(s-b)(s-c)} = \sqrt{12(12-5)(12-9)(12-10)} = \sqrt{12 \times 7 \times 3 \times 2}$.

Calculating that: $12 \times 7 = 84$, $84 \times 3 = 252$, $252 \times 2 = 504$. So, $\Delta = \sqrt{504}$.

Simplify $\sqrt{504}$. Let's factor 504: $504 \div 2 = 252$, $\div 2$ again $= 126$, $\div 2 = 63$, $\div 3 = 21$, $\div 3 = 7$.

So, $504 = 2^3 \times 3^2 \times 7$. Therefore, $\sqrt{504} = 2^{1.5} \times 3 \times \sqrt{7} = 2 \times \sqrt{2} \times 3 \times \sqrt{7} = 6\sqrt{14}$.

Wait, let me check that.

Wait, $\sqrt{504} = \sqrt{16 \times 31.5}$. Hmm, maybe I did that wrong. Wait, $504 \div 36 = 14$, so $\sqrt{504} = \sqrt{36 \times 14} = 6\sqrt{14}$. Yes, that's correct.

So, $\Delta = 6\sqrt{14}$.

Now, compute $R = \frac{abc}{4\Delta}$. Let's assign $a = BC = 9$, $b = AC = 10$, $c = AB = 5$.

So, $R = \frac{9 \times 10 \times 5}{4 \times 6\sqrt{14}} = \frac{450}{24\sqrt{14}}$.

... [continued]

---

**DeepSeek-R1-Distill-Qwen-7B (SFT)    AIME24 Problem #2**
**m=(172 vs.) 196 chrF=29.56 (Incorrect)**

Okay, so I've got this geometry problem here about triangle ABC inscribed in a circle $\omega$. The tangents at B and C meet at point D, and then AD intersects $\omega$ again at P. The sides AB, BC, and AC are given as 5, 9, and 10 respectively. I need to find the length of AP, express it as a reduced fraction m/n, and then find m + n. Hmm, okay, let me try to figure this out step by step.

First, I should probably sketch the problem to visualize it better. So, triangle ABC with sides AB=5, BC=9, AC=10. That seems like a scalene triangle. The circle $\omega$ is the circumcircle of ABC, which means all three vertices lie on $\omega$. The tangents at B and C intersect at D. So, D is the ex-pole of BC with respect to $\omega$? Or maybe it's just the ex-pole. Hmm, tangents from a common external point to a circle are equal in length, so DB = DC. That might be useful later.

Then, the line AD intersects $\omega$ again at P. So, AD is a secant line from A through D, intersecting the circle again at P. So, AP is the segment from A to P, which is part of AD. So, I need to find the length of AP.

Given the sides of the triangle, maybe I can use some properties of circles, power of a point, similar triangles, or something like that. Let me think.

Given that D is the intersection of the tangents at B and C, D is the ex-pole opposite A, right? Or maybe it's the ex-pole? Wait, actually, the point where two tangents meet is called the ex-pole if it's outside the circle, which it is because tangents are drawn from outside the circle.

So, point D is outside the circle $\omega$, and DB and DC are tangent to $\omega$. So, by the power of point D with respect to $\omega$, we have DB² = DC² = DA * DP. That might come in handy.

Power of a point says that for a point D outside a circle, the square of the tangent length is equal to the product of the lengths of the secant segments. So, in this case, DB² = DA * DP.

So, if I can find DB, then I can relate it to DA and DP. But I don't know DB yet. Maybe I can compute it somehow.

Alternatively, maybe I can use coordinate geometry. Assign coordinates to the points and compute everything. But that might get complicated. Let me see if I can find a more synthetic approach.

Wait, perhaps using triangle similarity or some theorem related to harmonic division or something.

Let me recall that in triangle ABC, with D being the ex-pole, AD is called the symmedian. So, AD is the symmedian of triangle ABC. That might be a useful property.

Symmedian reflects the median over the angle bisector. So, if I can find the median from A, then reflect it over the angle bisector to get the symmedian.

But I don't know if that's directly helpful here. Maybe using power of a point is still a better approach.

So, let's try to find DB. Since DB is the tangent from D to $\omega$, and D is the intersection of tangents at B and C. So, D is the ex-pole, and DB = DC.

... [continued]

