# OpenReview forum: "RL Squeezes, SFT Expands: A Comparative Study of Reasoning LLMs"
_ICLR.cc/2026/Conference — ICLR 2026 Poster_

### Official Review · Reviewer_mxYj · 2025-10-26

**Soundness:** 3
**Presentation:** 3
**Contribution:** 3
**Rating:** 4
**Confidence:** 4

**Summary:**

Summary:
This paper compares how reinforcement learning (RL) and supervised fine-tuning (SFT) shape reasoning in large language models. RL “squeezes” reasoning by concentrating paths, while SFT “expands” it by diversifying correct trajectories. Graph-based analyses explain why the SFT to RL pipeline works effectively.

**Strengths:**

The paper shows a novel analytic framework for The trajectory- and step-level “reasoning graph” approach provides a creative and rigorous lens for understanding LLM reasoning beyond metrics like Pass@k.

The work sheds light on why SFT+RL remains the dominant fine-tuning recipe for reasoning LLMs and offers potential implications for efficient data curation and model design.

**Weaknesses:**

The evaluated datasets (AIME24, AIME25, AMC23) are all from the same competition series and share similar problem distributions. This narrow scope makes it difficult to generalize the conclusions. For example, the claim that “applying SFT to the Base model increases the number of correct trajectories, showing that SFT teaches new solution strategies absent in the Base model” is not consistently significant, particularly for the AMC23 dataset and the 14B model.

The experiments use pretrained and post-trained models from the same family (e.g., Qwen and DeepSeek-based models). This raises concerns that the authors may not have controlled for shared training samples or overlapping data distributions, which could confound the observed “squeeze–expand” effects.

The paper omits several key implementation details required for replication. See question for details.

**Questions:**

The main findings rely heavily on clustering reasoning trajectories and sentences, yet the paper reports no clustering-quality metrics such as ARI, NMI, or Silhouette scores are reported. Without such validation, it is difficult to assess whether the clustering results genuinely reflect reasoning structure or are artifacts of algorithmic choices.


The paper adopts BGE-large-en-v1.5 (Xiao et al., 2024) for sentence embeddings instead of using the LLM’s own internal representations. What is the rationale for this choice? Given that the goal is to analyze reasoning behaviors within each model, it seems more natural to extract embeddings directly from the model under analysis, since external embeddings may not align with the model’s latent reasoning space.

The paper states that edge weights are defined by the Euclidean distance between cluster centroids, which suggests that every pair of nodes could be connected, effectively creating a dense graph. Could the authors clarify whether any thresholding, sparsification, or top-k selection was applied when constructing edges? If not, how should we interpret structural metrics such as degree, modularity, or centrality that assume sparse connectivity?

---

> ### Author Response · Authors · 2025-11-21
>
> ### Reviewr mxYj,
> We sincerely appreciate your detailed and constructive feedback.
>
> ### Beyond Mathematical Domains (W1)
> > This narrow scope makes it difficult to generalize the conclusions.
> > not consistently significant, particularly for the AMC23 dataset and the 14B model.
>
> We extend our experiments to code generation using HumanEval [1]. In the trajectory-level analysis (detailed in Appendix C.8), we observe that applying RL substantially reduces the number of incorrect trajectories (54.4 $\rightarrow$ 4.6 and 8.4 $\rightarrow$ 3.5), while applying SFT increases the number of correct trajectories (23.6 $\rightarrow$ 48.9) (Fig. 21). This behavior is fully consistent with the main results on math domain (Fig. 15). In the step-level analysis, we further analyze how RL and SFT affect the exponential decay rate $\beta$. Across visitation frequency, degree, and betweenness centrality (Fig. 37), RL consistently increases $\beta$, whereas SFT decreases it. In addition, the graph topology metrics (Fig. 38) exhibit trends that are consistent with the main results (Fig. 8).
>
> ### Overlapping Data (W2)
> > The experiments use pretrained and post-trained models from the same family (e.g., Qwen and DeepSeek-based models). This raises concerns that the authors may not have controlled for shared training samples or overlapping data distributions, which could confound the observed “squeeze–expand” effects.
>
> While the main results use Qwen and DeepSeek-distilled models, we have validated our findings, RL squeezes and SFT expands, with Llama-3.1-8B (student) and DeepSeek-R1 (teacher) in Appendix B.4 at the initial submission.
> These models differ substantially in architecture, training pipelines, and pretraining corpora, which mitigates the concern that our observations arise from shared training samples or overlapping data distributions.
> Given that the same qualitative pattern emerges across these heterogeneous model families, we believe the squeeze–expand effect is robust to differences in data sources and pretraining distributions.
>
> ### Clustering Method (Q1)
> > it is difficult to assess whether the clustering results genuinely reflect reasoning structure.
>
> Since there is no ground-truth or canonical baseline for discretizing the continuous sentence space into a graph, we perform an ablation study by varying the graph construction parameters (detailed in Appendix D.4). We vary the graph construction procedure by changing the number of clusters to $K=1000, 3000$, measuring distances in cosine space, and using a different family of sentence embeddings (GTE). Using these settings, we construct reasoning graphs on AIME24 with the 7B models and estimate the exponential decay rate $\beta$ of the rank plots. Across all metrics, visitation frequency, degree, and betweenness centrality, and for all parameter choices, we observe the same qualitative pattern as in the main results: RL consistently increases, whereas SFT decreases $\beta$ (Fig. 32). These results demonstrate that our findings are robust to the graph-construction choices, and not artifacts of a particular clustering configuration.

---

> ### Author Response · Authors · 2025-11-21
>
> ### The Use of Sentence Embedding (Q2)
> > The paper adopts BGE-large-en-v1.5 (Xiao et al., 2024) for sentence embeddings instead of using the LLM’s own internal representations. What is the rationale for this choice?
>
> We deliberately use sentence embeddings rather than internal representations to enable fair comparison of reasoning graphs across models (Base, RL, SFT, SFT+RL) in a unified semantic space. Using each model's internal representations would result in comparing graphs in different representation spaces. To make this point clearer, we have added this point in manuscript.
>
> ### Sparsifying Reasoning Graphs (Q3)
> > Could the authors clarify whether any thresholding, sparsification, or top-k selection was applied when constructing edges?
>
> Thank you for raising this sharp point. Our reasoning graphs are not dense (edge density around 0.05, at most 0.1), as shown in Fig. 8, and far from complete graphs. Additionally, based on the assumption that reasoning steps are semantically connected, we avoid disconnected graphs. For instance, when integrating $M=256$ reasoning generations for each problem, we first ensure connectivity by adding the problem $x$ as the initial node (lines 227-232).
> To confirm the robustness of our approach and address your concern regarding clustering-based node definitions, we have performed additional sparsification experiments (detailed in Appendix D.5). We apply a sparsification procedure to reasoning graphs of 7B models based on sentence embeddings. Specifically, for each node, we retain only the top-$k$ nearest connected nodes in terms of the L2 distance between sentence embeddings, and prune all other edges. We consider both $k=10, 20$. For these sparsified graphs, we estimate the exponential decay rate $\beta$ of the degree and centrality of the rank plots on the AIME24, AIME25, and AMC23 domains (Fig. 33). We obtain the same qualitative behavior as in the main results without sparsification, where RL increases $\beta$, while SFT decreases $\beta$. Furthermore, when we compute eight topological metrics (Fig. 34), we observe trends that are fully consistent with our main results (Fig. 8).
>
> We sincerely appreciate your insightful comments. Please let us know if our responses resolve your concerns or if additional clarification is required.
>
> ---
> References
>
> [1] Evaluating Large Language Models Trained on Code. arXiv preprint. 2107.03374.

---

> > ### Author Response · Authors · 2025-11-28
> > **Gentle Reminder: Rebuttal to Reviewer mxYj**
> >
> > Thank you again for your thoughtful and constructive feedback on our paper. We wanted to follow up on our previous response, in which we have carefully addressed each of your concerns. In particular, we have conducted experiments on code generation (beyond math), clarified that overlapping data is considered in our Llama experiments, performed additional experiments with different clustering methods and sentence embeddings, and tested graph sparsification. All of them showed consistent results.. These revisions have been reflected in the updated manuscript.
> >
> > We would be very grateful if you could take a moment to review our responses and the revised paper at your convenience. We believe these changes have substantially improved the work and would greatly appreciate your reconsideration of the evaluation.

---

### Official Review · Reviewer_pCKm · 2025-10-27

**Soundness:** 3
**Presentation:** 3
**Contribution:** 3
**Rating:** 4
**Confidence:** 4

**Summary:**

This work provides a novel framework to analyze the properties of two post-training techniques of LLM - SFT and RL.
It investigates the reasoning processes of base, SFTed, RLed and SFTed+RLed models at two levels of granularity: the trajectory-level which examines complete reasoning outputs, and the step-level, which analyzes reasoning graphs whose nodes correspond to individual reasoning steps.

Based on the analysis existing models, covering Qwen2.5-Math-1.5B, Qwen2.5-Math-7B and Qwen2.5-14B, the authors get the conclusion that RL compresses incorrect trajectories, whereas SFT expands correct ones (trajectory-level); RL concentrates reasoning functionality into a small subset of steps, while SFT homogenizes it across many steps (step-level).

**Strengths:**

1. This work proposes a novel and solid analysis framework based on reasoning paths to better understand the effects of SFT and RL in LLM post-training, particularly from the perspective of reasoning diversity (e.g., pass@k vs. pass@1).

2. The framework design is rigorous and well-grounded in theory.

3. The paper is clearly written and well-presented, making the methodology and findings easy to follow.

4. The experimental analysis is comprehensive and covers multiple model scales and post-training settings.

**Weaknesses:**

## Representativeness of the SFT Models

I am concerned about whether the models chosen in this paper accurately represent conventional SFT.

The authors use the DeepSeek-R1-Distill-Qwen series as SFT models; however, these models are trained via knowledge distillation (KD), which differs from standard SFT practices. In KD, for each prompt, multiple responses from teacher models are used for supervision (as described in Sec. 2.3.3 of Guo et al., 2025). This setup might make properties diverge from a pure SFT setting and potentially enhance the diversities by incorporating richer COT paths. (In fact, it is somehow close to offline RL).

Therefore, the behavior of KD-based models may not truly reflect that of conventional SFT models. If this is the case, some of the conclusions in the paper may not fully hold.




--------------_

- Guo, D., Yang, D., Zhang, H., Song, J., Zhang, R., Xu, R., ... & He, Y. (2025). Deepseek-r1: Incentivizing reasoning capability in llms via reinforcement learning. arXiv preprint arXiv:2501.12948.

**Questions:**

1. Do the properties and construction methods of SFT datasets influence the resulting SFT models? If so, could this affect the validity or generality of the paper’s conclusions?

---

> ### Author Response · Authors · 2025-11-21
>
> ### Reviewr pCKm,
> We truly appreciate your valuable and insightful feedback, which improves our work.
>
> ### Representativeness of SFT (W&Q)
> > Therefore, the behavior of KD-based models may not truly reflect that of conventional SFT models.
>
> > Do the properties and construction methods of SFT datasets influence the resulting SFT models?
>
> Our main results utilized DeepSeek-R1-Distill-Qwen series as SFT models following [1], and we have conducted additional experiments to further validate that our conclusions hold more broadly.
>
> We applied standard SFT to the Qwen2.5-Math-1.5B using the high-quality filtered s1k-1.1 dataset [2] for 5 epochs, and conducted trajectory-level analysis (detailed in Appendix C.7) and step-level analysis (detailed in Appendix D.6) using Qwen2.5-Math-1.5B (Base), Qwen2.5-Math-1.5B-Oat-Zero (RL), and Qwen2.5-Math-1.5B_s1k-1.1 (SFT). In the trajectory-level experiments, we consistently observe across AIME24, AIME25, and AMC23 that RL reduces the number of incorrect trajectories (88.2 $\rightarrow$ 46.1 in AIME24, 97.1 $\rightarrow$ 41.1 in AIME25, and 52.3 $\rightarrow$ 21.0 in AMC23) while increasing correct  (4.4 $\rightarrow$ 6.9 in AIME24, 4.4 $\rightarrow$ 8.3 in AIME25, and 15.1$\rightarrow$ 27.6 in AMC23). In the step-level experiments, the exponential decay rate β consistently increased under RL (0.035 $\rightarrow$ 0.085 for visitation frequency, 0.020 $\rightarrow$ 0.041 for degree, and 0.054 $\rightarrow$ 0.128 for betweenness centrality) and decreased under SFT (0.035 $\rightarrow$ 0.023 for visitation frequency, 0.020 $\rightarrow$ 0.014 for degree, and 0.054 $\rightarrow$ 0.035 for betweenness centrality) in AIME24, AIM25, and AMC23 domains. Furthermore, examining graph topology metrics (Fig. 36), we observe results consistent with the main results (Fig. 8) across all eight metrics. These results support that the "expand" effect of SFT persists even under standard SFT with single response per problem. We have added this discussion to the manuscript.
>
> We sincerely appreciate your insightful comments. Please let us know if our responses resolve your concerns or if additional clarification is required.
>
> ---
> References
>
> [1] Yue et al. Does Reinforcement Learning Really Incentivize Reasoning Capacity in LLMs Beyond the Base Model? NIPS 2025, Oral.
>
> [2] Muennighoff et al. s1: Simple test-time scaling. EMNLP 2025.

---

> > ### Comment · Reviewer_pCKm · 2025-11-27
> >
> > Thank you for the detailed response.
> >
> > I think the response addresses my concern and I will raise the score to 6.

---

> > > ### Author Response · Authors · 2025-12-03
> > >
> > > Thank you for your constructive feedback and positive evaluation. We greatly appreciate your time and suggestions. Your feedback has helped us improve our work.

---

### Official Review · Reviewer_YntV · 2025-10-29

**Soundness:** 3
**Presentation:** 3
**Contribution:** 3
**Rating:** 6
**Confidence:** 4

**Summary:**

This paper analyzes how SFT and RL on Verifiable Rewards shape the LLM reasoning process. They perform a two-level analysis: (1) trajectory-level clustering of sampled solutions, and (2) step-level reasoning graphs built from sentence embeddings of the models' CoT. The core findings are the following: (1) RL compresses incorrect trajectories while SFT expands correct trajectories, helping explain why SFT --> RL is the prevailing training recipe; (2) at step level, RL also consolidates reasoning graph functionality into fewer steps. Experiments are shown for 1.5B/7B/14B models on AIME24/25 and AMC23 with controlled sampling.

**Strengths:**

1. It presents a timely analysis for a RL/SFT fine-tuning on LLMs, which helps explaining and understanding  the use of these techniques for fine-tuning.
2. The authors support their insights with a solid set of experiments which include multiple models.
3. The authors perform a detailed analysis at the whole reasoning trajectory, and at the step level combined with fundament computer science analysis based on graphs.

**Weaknesses:**

1. Reasoning graphs depend on (a) sentence segmentation, (b) embedding model, (c) k=2000, and (d) Euclidean distances. Some ablation on of these points can help strengthen the derived insight. Some of these ablations should be easy to run.
2. Local graphlet proportions can look similar across models while accuracies differ; more analysis is needed to connect which topological features track Pass@1/Pass@k
3.  In the graph analysis, step-level graphs aggregate all sampled responses for a problem without separating by correctness, so topology may conflate successful and failing behaviors.  Similarly, edge distance is Euclidean between K-means centroids from a single embedding model; no evidence is given that this correlates with semantic step difference, and cosine/other spaces aren’t tested.

**Questions:**

1. Any preliminary results on code or scientific reasoning to test whether RL still weakens community structure and SFT boosts global efficiency?
2. I'd suggest the Figure should provide some insight, and the caption should explain it. For instance, what's the value of the 4 versions of the same plot in Figure 4? Or similarly, what is the value of each metric in Figure 7?
3. The findings in the paper are interesting, but the authors fail to point out how they are useful for improving them, do you have some comments on how we can leverage these insights?

---

> ### Author Response · Authors · 2025-11-21
>
> ### Reviewr YntV,
>
> We thank you for pointing out the issue with the figure captions. We have improved the captions for Fig.4&7 (Fig. 5&8 after revision) (Q2).
>
> ### Other Domains (Q1)
> > code or scientific reasoning to test whether RL still weakens community structure and SFT boosts global efficiency?
>
> Yes, RL still weakens community structure and SFT boosts global efficiency in the code domain. Using HumanEval [1], we conducted both trajectory-level (detailed in Appendix C.8) and step-level analyses (detailed in Appendix D.7), and in all cases we obtained results consistent with those in the math domain. In particular, for the global topological metrics, the results in the code domain (Fig. 38) exhibited the same trends as those in the math domain (Fig. 8) across all eight metrics.
>
> ### Ablations on Graph Construction (W1&W3)
> > Reasoning graphs depend on (a) sentence segmentation, (b) embedding model, (c) k=2000, and (d) Euclidean distances. Some ablation on of these points can help strengthen the derived insight
>
> > Euclidean between K-means centroids from a single embedding model; no evidence is given that this correlates with semantic step difference, and cosine/other spaces aren’t tested.
>
> To examine the robustness of our reasoning graph construction, we conducted experiments with the following variations: (i) varying the cluster count to $K=1000, 3000$, (ii) measuring distances in cosine space, and (iii) using a different sentence embedding model, GTE-base-en-v1.5 [2]. Across all settings, the exponential decay rate $\beta$ of the rank plots consistently increased under RL and decreased under SFT (Fig. 32). These results demonstrate that our main results hold regardless of the specific graph construction setting.
>
> ### Graph Topology and Pass@$k$ (W2)
> > more analysis is needed to connect which topological features track Pass@1/Pass@k
>
> To investigate the relationship between topological features and Pass@$1$/Pass@$k$, we computed Pass@$1$ / Pass@$k$ on AIME24, AIME25, and AMC23 (Fig. 30) and compared them with the topological metrics (Fig. 31). The results reveal that Pass@$1$ / Pass@$k$, the ability to effectively explore the solution space and reach the correct answer in a single attempt, correlates positively with *global efficiency* and *algebraic connectivity*, while correlating negatively with *modularity*. This suggests that graph structures critical for reasoning capability do not exhibit clustered structures like high modularity, but rather possess structures that enable efficient traversal across the entire graph. Details are provided in Appendix D.3.
>
>
> ### Implications and Insights (Q3)
> >  do you have some comments on how we can leverage these insights?
>
> As noted above, graph structures that facilitate traversal across the entire graph without high modularity are critical for reasoning.
> Potential directions include selective process reward allocation in RL, targeting functionally important nodes (e.g., high degree, centrality) rather than all steps, and topology-based data curation to classify and filter reasoning traces (lines 416-422).
>
> We sincerely appreciate your insightful comments. Please let us know if our responses resolve your concerns or if additional clarification is required.
>
> ---
> References
>
> [1] Evaluating Large Language Models Trained on Code. arXiv preprint. 2107.03374.
>
> [2] mGTE: Generalized Long-Context Text Representation and Reranking Models for Multilingual Text Retrieval. ACL 2024.

---

> > ### Comment · Reviewer_YntV · 2025-11-27
> >
> > Thank you for your comments and added results. It helps me understand some of my concerns. Your paper tries to study the behavior of reasoning with SFT and RL and the behavior needs to be clearly analyzed in order to not make false claims.
> >
> > I think these new results are interesting, but they need to be coherently included into the paper results, instead of just pointing to the appendices. Some of these results are relevant to understand the mechanisms analyzed in the study. For instance, results from other domains (e.g. code) or the relation of the graph topology with the performance results,
> >
> > Yet, I believe this paper is needs to explain or show how to leverage these insights or how they can be used to improve our current methods.
> >
> > My score represents my current thoughts about the paper.

---

> > > ### Author Response · Authors · 2025-12-03
> > >
> > > Thank you very much for your thoughtful feedback. It has helped us improve our work.
> > > We have incorporated the additional experimental results (including those on code) into the main paper. The insights drawn from these results have been added to the discussion section.
> > >
> > > We sincerely appreciate your comments and the opportunity to refine our study.

---

### Official Review · Reviewer_oB2v · 2025-11-04

**Soundness:** 2
**Presentation:** 3
**Contribution:** 2
**Rating:** 4
**Confidence:** 3

**Summary:**

The authors examine how two training strategies, reinforcement learning with verifiable rewards (RLVR) and supervised fine-tuning on reasoning traces (SFT), reshape LLM reasoning. Their central result: RLVR compresses the reasoning space, reducing path diversity and concentrating effort on a small number of high-impact steps, whereas SFT expands it, introducing many correct solution paths and distributing functionality across more steps. This contrast clarifies why the standard SFT + RL pipeline succeeds: SFT first generates a richer set of good paths; RL then trims the weak ones to raise Pass@1.

**Strengths:**

1. The trajectory and step level analysis for both SFT and RL are interesting to me.
Trajectory clustering shows RL compresses incorrect paths while SFT expands correct ones, motivating the common two-stage recipe (SFT then RL).

2. The paper builds “reasoning graphs” and profiles them with edge density, modularity, assortativity, centralization, global efficiency, etc., revealing RL concentrates functionality into hubs while SFT distributes it.

3. The paper is easy to follow.

**Weaknesses:**

1. Even though the analysis is interesting, the paper’s scope and novelty are not sufficient for ICLR.

2. The domain and model scope are narrow. The evidence centers on verifiable math datasets and a limited set of public checkpoints, leaving generalization to coding/science tasks and broader architectures untested.

**Questions:**

1. Can we test on other domain datasets to see if the observed effects replicate?

---

> ### Author Response · Authors · 2025-11-21
>
> ### Reviewer oB2v,
> We greatly appreciate your critical feedback.
>
> ### Scope and Novelty (W1)
> >Even though the analysis is interesting, the paper’s scope and novelty are not sufficient for ICLR.
>
> Our work addresses a critical and timely question that is of significant interest to the community: understanding the principled differences between SFT (imitation) and RL (reinforcement) in LLM post-training. We recognize that the community has shown substantial interest in the line of work on **pass@k** results [1], "**SFT memorizes RL generalizes**" [2], and the practical success of **SFT+RL** pipelines (e.g., [3][4]). Our work provides novel insights into these phenomena through the lens of rigorous reasoning paths, while many works rely on just accuracy metrics. Our trajectory and step-level analysis demonstrates that the "squeeze" effect improves Pass@1 and "expand" effect elevates Pass@k (k>>1). The squeezing and expanding of graph functionality in RL and SFT parallel RL's generalization and SFT's memorization. RL's compression of incorrect and SFT's expansion of correct trajectories explain SFT-then-RL procedure. Additionally, step-level (semantic) RL's "squeeze" effect, where RL **concentrates functionality** (e.g., hub, central node) into a subset of nodes while diminishing the functionality of others, aligns with the widely discussed results on token-level entropy separation after RL, where the entropy of a subset of “thinking” tokens increases while that of other tokens decreases [5] (line 403-407). We believe these results represent significant contributions.
>
> ### Experimental Setting (W2)
> > The evidence centers on verifiable math datasets and a limited set of public checkpoints
>
> We chose to use publicly available checkpoints whenever possible, following prior work ([1] (NISP2025 Oral) [6] (NIPS2025)), to ensure reproducibility with widely used models. To address your concern, we have now conducted additional experiments in Appendix C.7 & D.6 with models we trained ourselves. We performed SFT on Qwen2.5-Math-1.5B with s1k-1.1 dataset [7] and In the trajectory-level experiments, we consistently observe across AIME24, AIME25, and AMC23 that RL reduces the number of incorrect trajectories (88.2 $\rightarrow$ 46.1 in AIME24, 97.1 $\rightarrow$ 41.1 in AIME25, and 52.3 $\rightarrow$ 21.0 in AMC23) while increasing correct  (4.4 $\rightarrow$ 6.9 in AIME24, 4.4 $\rightarrow$ 8.3 in AIME25, and 15.1$\rightarrow$ 27.6 in AMC23). In the step-level experiments, the exponential decay rate β consistently increased under RL (0.035 $\rightarrow$ 0.085 for visitation frequency, 0.020 $\rightarrow$ 0.041 for degree, and 0.054 $\rightarrow$ 0.128 for betweenness centrality) and decreased under SFT (0.035 $\rightarrow$ 0.023 for visitation frequency, 0.020 $\rightarrow$ 0.014 for degree, and 0.054 $\rightarrow$ 0.035 for betweenness centrality) in AIME24, AIM25, and AMC23 domains. Furthermore, examining graph topology metrics (Fig. 36), we observe results consistent with the main results (Fig. 8) across all eight metrics.
>
> These additional experiments with our own trained models directly address your concern that our results were only validated on public checkpoints, and they further strengthen the overall claims of our paper.

---

> ### Author Response · Authors · 2025-11-21
>
> ### Other Domains (W2&Q1)
> > Can we test on other domain datasets to see if the observed effects replicate?
>
> Yes, we were able to replicate the results even in the code domains. To validate our results beyond the math domain, we extend our experiments to code generation using HumanEval [8]. In the trajectory-level analysis (detailed in Appendix C.8), we observe that applying RL substantially reduces the number of incorrect trajectories (54.4 $\rightarrow$ 4.6 and 8.4 $\rightarrow$ 3.5), while applying SFT increases the number of correct trajectories (23.6 $\rightarrow$ 48.9) (Fig. 21). This behavior is fully consistent with the main results on math domain (Fig. 15). In the step-level analysis, we further analyze how RL and SFT affect the exponential decay rate $\beta$. Across visitation frequency, degree, and betweenness centrality (Fig. 37), RL consistently increases $\beta$, whereas SFT decreases it. In addition, the graph topology metrics (Fig. 38) exhibit trends that are consistent with the main results (Fig. 8).
>
> We sincerely appreciate your insightful comments. Please let us know if our responses resolve your concerns or if additional clarification is required.
>
> ---
> References
>
> [1] Yue et al. Does Reinforcement Learning Really Incentivize Reasoning Capacity in LLMs Beyond the Base Model? NIPS 2025, Oral.
>
> [2] Chu et al. SFT Memorizes, RL Generalizes: A Comparative Study of Foundation Model Post-training. ICML 2025.
>
> [3] Guo et al. DeepSeek-R1: Incentivizing Reasoning Capability in LLMs via Reinforcement Learning. 2025.
>
> [4] Chen et al. AceReason-Nemotron: Advancing Math and Code Reasoning through Reinforcement Learning. 2025.
>
> [5] Beyond the 80/20 Rule: High-Entropy Minority Tokens Drive Effective Reinforcement Learning for LLM Reasoning. 2025.
>
> [6] Minegishi et al. Topology of Reasoning: Understanding Large Reasoning Models through Reasoning Graph Properties. NIPS 2025.
>
> [7] Muennighoff et al. s1: Simple test-time scaling. EMNLP 2025.
>
> [8] Evaluating Large Language Models Trained on Code. arXiv preprint. 2107.03374.

---

> > ### Author Response · Authors · 2025-11-28
> > **Gentle Reminder: Rebuttal to Reviewer oB2v**
> >
> > Thank you again for your thoughtful and constructive feedback on our paper. We wanted to follow up on our previous response, in which we have carefully addressed each of your concerns. In particular, we have clarified our novelty and contribution, conducted experiments with self-trained models, and validated our findings on another domain (code), all showing consistent results. These revisions have been reflected in the updated manuscript.
> >
> > We would be very grateful if you could take a moment to review our responses and the revised paper at your convenience. We believe these changes have substantially improved the work and would greatly appreciate your reconsideration of the evaluation.

---

### Author Response · Authors · 2025-11-21
**Summary of Rebuttal and Discussion**

### Dear PCs, SACs, ACs, and Reviewers,

We wish to express our strong appreciation to the reviewers for their insightful comments on our paper. We feel the feedbacks have helped us significantly improve the paper. To assist the new AC, we summarize the discussion with the reviewers. We received responses from **2/4** reviewers before the score reversion, but unfortunately the remaining 2 reviewers were not able to respond.

### Strength
The reviewers recognized
- our trajectory and step-level reasoning graph approach offers a new perspective for analyzing LLM reasoning beyond Pass@k metrics.
- Our analysis explains why the widely adopted SFT+RL recipe is so effective in practice.
- Our analysis is supported by multiple models and fundamental computer science analysis based on graphs.
- The paper is clearly written and well-presented.

### Concerns
The reviewers pointed out the following issues.
| Issue | Review | Response / Revision |
|---|--------|-------------|
| Domain | oB2v(W2&Q1), YntV(Q1), mxYj(W1) | Conducted **additional experiments** |
|Graph construction | YntV(W1), mxYj(Q1) | Conducted **additional experiments** |
| Sentence embedding | YntV(W1&W3), mxYj(Q2) | Conducted **additional experiments** and clarified in the response |
| Representativeness of SFT | oB2v(W2), pCKm(W&Q)  | Conducted **additional experiments** |
| Novelty and Scope | oB2v(W1) | Clarified in the response and revised the manuscript |
| Implications | YntV(Q3) | Clarified in the response and revised the manuscript |
| Model family | mxYj (W2) |  Clarified in the response and revised the manuscript  |
| Topology and pass@k | YntV (W2) | Conducted **additional experiments** |
| Graph sparsification | mxYj(Q3) | Conducted **additional experiments** |
| Figure caption | YntV(Q2) | Revised the manuscript |

### Summary of Additional Experiments
- **Appendix C.5**: Trajectory-level analysis on the models we performed SFT on (Reviewer **pCKm**)
- **Appendix C.6**: Trajectory-level analysis on code domain (Reviewer **oB2v**, **YntV**, **mxYj**)
- **Appendix D.3**: Analysis of the relationship between Pass@$k$ and graph topology
- **Appendix D.4**: Ablation study on reasoning graph constructions (Reviewer **YntV**, **mxYj**)
- **Appendix D.5**: Step-level analysis on sparsified graphs (Reviewer **mxYj**)
- **Appendix D.6**: Step-level analysis on the models we performed SFT on (Reviewer **pCKm**)
- **Appendix D.7**: Step-level analysis on code domain (Reviewer **oB2v**, **YntV**, **mxYj**)

Through these additional experiments, (i) we confirmed that our trajectory and step-level findings, "RL squeezes and SFT expands", hold not only in the math domain but also in the code domain. (ii) We reproduced the results with models trained from scratch via SFT, (iii) performed ablation studies on graph construction parameters and sentence embedding to verify consistent results across different graph configurations, and (iv) confirmed that the results remain consistent even with sparsified graphs.

The added content is highlighted in **red** in the manuscript.

All in all, we again sincerely thank all reviewers for their feedback. We believe we have thoroughly addressed all concerns through additional experiments and revisions, and we hope our responses will be helpful to the new AC in the revised review process.

Sincerely, Authors

---

### Meta-Review · Area_Chair_JPCX · 2026-01-15

**Summary:**

Reviewers believed that this paper gives a timely and well-presented analysis to understanding post-training effects beyond performance. The authors use trajectory-level clustering of the sampling reasoning path and step-level "reasoning graphs", which establish a novel analysis framework.

The concerns mainly focus on:
1. Novelty and scope for ICLR. Reviewer (oB2v) viewed the analysis as interesting but questioned whether the incremental novelty and overall scope rose to ICLR’s bar
2. Generality across domains and tasks. Multiple reviewers considered that the initial evidence leaned heavily on verifiable math benchmarks (AIME), with requests to validate whether the “RL squeezes / SFT expands” pattern holds in other domains (e.g., code, science).
3. Methodological robustness and validity: YntV and mxYj raised questions about sensitivity to sentence segmentation, embedding choice, clustering hyperparameters and distance metrics.

**Reviewer Concerns:**

Addressed by the rebuttal and revision:
The concerns on domain generalization, representativeness of SFT, robustness to graph construction, and presentation issues have been addressed.

Partially addressed:
1. Causality and correlation: The added topology–performance analysis appears correlational,  the paper still does not establish that modifying topology causes the performance shifts, or being a byproduct.
2. Math + HumanEval is a meaningful expansion, but “science reasoning / broader tasks”  remain limited, thus the generality claim should still be phrased cautiously.

**Reviewer Scores:**

Reviewer pCKm: raised from 4 to 6

Reviewer YntV: score 6. As the authors has addressed most of the issues, I would expect the score will maintain 6 or increase to 7.

Reviewer pCKm: score 4. The rebuttal directly targets these concerns with code-domain results, Llama-family evidence and abaltions. I would expect 4 to 6 if they had engaged and accepted the new evidence.

Reviewer oB2v: score 4. Their main concerns was ICLR-level novelty/scope plus generalization beyond math. I would expect their score will maintain 6.

---

### Decision · Program_Chairs · 2026-01-26

Accept (Poster)